# ENDOWING VISUAL REPROGRAMMING WITH ADVERSARIAL ROBUSTNESS

**Shengjie Zhou**[1*]  **Xin Cheng**[1*]  **Haiyang Xu**[2]  **Ming Yan**[2]
**Tao Xiang**[1]  **Feng Liu**[3]  **Lei Feng**[4,5†]
[1]Chongqing University  [2]Alibaba Group  [3]University of Melbourne
[4]Southeast University  [5]Idealism Technology (Beijing)

## ABSTRACT

Visual reprogramming (VR) leverages well-developed pre-trained models (e.g., a pre-trained classifier on ImageNet) to tackle target tasks (e.g., a traffic sign recognition task), without the need for training from scratch. Despite the effectiveness of previous VR methods, all of them did not consider the *adversarial robustness* of reprogrammed models against adversarial attacks, which could lead to unpredictable problems in safety-crucial target tasks. In this paper, we empirically find that *reprogramming pre-trained models with adversarial robustness and incorporating adversarial samples from the target task during reprogramming can both improve the adversarial robustness of reprogrammed models*. Furthermore, we propose a theoretically guaranteed adversarial robustness risk upper bound for VR, which validates our empirical findings and could provide a theoretical foundation for future research. Extensive experiments demonstrate that by adopting the strategies revealed in our empirical findings, the adversarial robustness of reprogrammed models can be enhanced.

## 1 INTRODUCTION

Visual pre-trained models (Krizhevsky et al., 2012; Russakovsky et al., 2015; Huang et al., 2017; Radford et al., 2021) obtained from the (usually large) source domain serve as a powerful foundation for a variety of visual target tasks. When faced with the relevant downstream tasks, the pre-trained models can be adapted to solve the tasks through fine-tuning. However, a significant limitation of fine-tuning methods (Pan et al., 2011; Ghifary et al., 2014; Kumar et al., 2022) is their requirement for partial or complete modification of model parameters, resulting in high training and storage costs, particularly when using with large models. In contrast, visual reprogramming (VR) (Lee et al., 2020; Kloberdanz et al., 2021; Chen et al., 2021; Neekhara et al., 2022; Wang et al., 2022; Chen, 2022) provides an alternative and efficient way to re-purposing the well-developed pre-trained models from source domains. Specifically, VR does not require modifications to the parameters of the pre-trained model. Instead, as illustrated in Figure 1, it fixes the pre-trained model and incorporates an *input transformation* (Yang et al., 2021; Bahng et al., 2022; Cai et al., 2024; Tsao et al., 2024) and an *output label mapping* (Elsayed et al., 2018; Tsai et al., 2020; Chen et al., 2023b) at the input and output stages, respectively, to adapt to the requirements of the target domain. Moreover, compared with general prompting methods (Jia et al., 2022; Cheng et al., 2023) in visual tasks, VR offers a model-agnostic approach that preserves the visual essence of the input images (Cai et al., 2024). Previous research has proposed numerous VR methods, which have achieved satisfactory results across a wide range of tasks, demonstrating the considerable potential of VR.

Despite many previous studies on VR, they have not considered the *adversarial robustness* (Szegedy, 2013; Ashmore et al., 2021) of the reprogrammed models in target domains, which is an important standard for measuring the quality of learned models, especially in safety-crucial domains like medical diagnosis (Hamid et al., 2017; Kadampur & Al Riyaee, 2020), autonomous driving (Grigorescu et al., 2020; Feng et al., 2021), and criminal justice (Zhong et al., 2018; Chalkidis et al., 2019).

---

[*]These authors contributed equally to this work.
[†]Corresponding author.

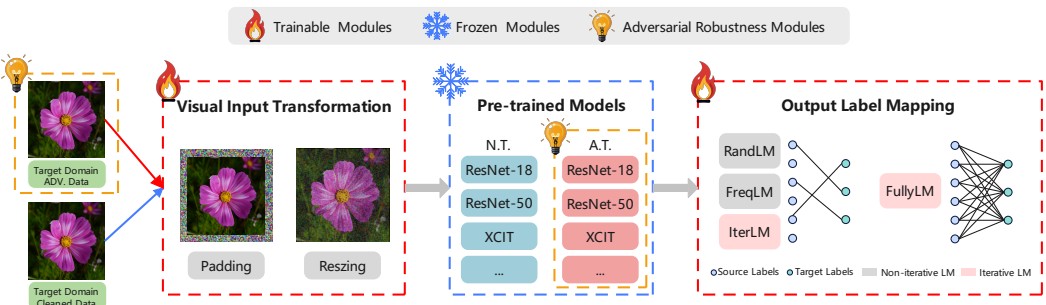

Figure 1: Overview and key highlights of VR with adversarial robustness. The main components are: **Training Data**, which comes from supervised data of the target domain task, including the cleaned data and generated adversarial data; **Visual Input Transformation**, which transforms data from the target domain to fit the inputs of the pre-trained model, including both Padding and Resizing methods; **Pre-trained Models**, which are well-developed classifiers in the source domain, including both naturally trained models (N.T.) that lack adversarial robustness and adversarially trained models (A.T.) that possess adversarial robustness; and **Output Label Mapping**, which maps labels from the source domain to the target domain, including one-to-one mapping (e.g., RandLM, FreqLM and IterLM) and fully mapping (FullyLM). We indicate that selecting adversarial data for reprogramming and choosing adversarially pre-trained models can both improve the adversarial robustness of the reprogrammed model.

Adversarial robustness refers to the ability of a model to maintain its performance under adversarial attacks (Goodfellow et al., 2014; Carlini & Wagner, 2017; Yao et al., 2019) which aim to add small perturbations that cause incorrect predictions. Numerous studies (Szegedy, 2013; Biggio et al., 2013) have demonstrated that naturally trained models exhibit significant vulnerabilities when confronted with adversarial attacks. However, it remains unclear whether the same phenomenon occurs in VR. Unfortunately, as shown in Figure 2, we empirically find that the reprogramming of naturally pre-trained models under various VR results in virtually no adversarial robustness on GTSRB. Notably, C-AVP (Chen et al., 2023a) focuses on improving the robustness of naturally pre-trained models within the same task, whereas it does not investigate the adaptation of a pre-trained model to achieve robustness across distinct tasks. Therefore, ensuring adversarial robustness is essential for the effective development of VR.

In this work, we for the first time investigate the issue of adversarial robustness within the context of VR. Our empirical findings reveal that while reprogramming a naturally pre-trained model can achieve high accuracy, its performance on adversarial examples is nearly zero. In contrast, reprogramming an adversarially pre-trained model yields a certain degree of adversarial robustness in the target domain, indicating that current reprogramming methods can partially leverage the adversarial robustness of the adversarially pre-trained models. Inspired by adversarial training (e.g., PGD-AT (Madry et al., 2018) and TRADES (Zhang et al., 2019a)), we incorporate adversarial examples from the target domain during the reprogramming process, which enhances the adversarial robustness of reprogrammed models, particularly yielding significant improvements when using an adversarially pre-trained model. For example, in Figure 2 , the adversarial robustness of either reprogramming an adversarially pre-trained ResNet-50 with natural reprogramming or reprogramming a naturally pre-trained ResNet-50 with adversarial reprogramming is higher than the original, and the adversarial robustness of reprogramming an adversarially pre-trained ResNet-50 with adversarial reprogramming is even better. These strategies are illustrated in Figure 1.

Furthermore, in order to theoretically understand the adversarial robustness of VR, we present a theoretically guaranteed adversarial robustness risk upper bound for VR. Specifically, our theoretical analysis establishes a connection between the target domain and the source domain, demonstrating that the adversarial robustness of VR is primarily influenced by these factors: 1) the adversarial robustness of the pre-trained model in the source domain; 2) the adversarial margin disparity discrepancy between the source and target domains; 3) the adversarial marginal risk in the target domain for the optimal model with adversarial robustness of source domain. This indicates that replacing a

pre-trained model lacking adversarial robustness with one that possesses adversarial robustness, as well as incorporating adversarial examples from target domain during reprogramming, can result in a reduced upper bound of adversarial risk, thereby validating our findings. Our main contributions can be summarized as follows:

Our main contributions can be summarized as follows:

- This study is the first to explore adversarial robustness within the context of VR, demonstrating the vulnerability of existing VR methods in the face of adversarial attacks.

- Our empirical results indicate that reprogramming pre-trained models with adversarial robustness and incorporating adversarial examples from the target task during the reprogramming can both improve the adversarial robustness of VR.

- We provide an adversarial risk upper bound for VR, which validates our empirical findings and provides a theoretical foundation for future research.

## 2 PRELIMINARIES

In this section, we introduce preliminary knowledge of the adversarial training and the visual reprogramming, as well as the notations we used throughout this paper.

### 2.1 ADVERSARIAL TRAINING

**Natural Training (N.T.).** Let us consider a $k$-class classification task with the input data $(\boldsymbol{x}, y) \in \mathbb{R}^d \times [k]$ sampled from an unknown data distribution with probability density $p(\boldsymbol{x}, y)$. Most existing learning algorithms for such classification tasks optimize the natural objective loss $\ell_{\mathrm{NT}}$ (e.g., cross-entropy loss) on clean examples to derive a well performing classifier $f : \mathbb{R}^d \to \mathbb{R}^k$ with parameters $\boldsymbol{\theta}$. The goal of N.T. can be expressed as:

$$\min_{\boldsymbol{\theta}} \mathbb{E}_{p(\boldsymbol{x}, y)} \left[ \ell_{\mathrm{NT}} \left( f(\boldsymbol{x} \mid \boldsymbol{\theta}), y \right) \right]. \tag{1}$$

However, previous studies (Szegedy, 2013; Biggio et al., 2013) have demonstrated that despite the high clean accuracy of naturally trained models on unperturbed examples, their accuracy significantly deteriorates when exposed to examples that have been subtly perturbed by adversaries (i.e., adversarial examples). These perturbations are so small that they are invisible to the human eye. We refer to the performance on adversarial examples as *adversarial robustness*. For simplicity, we denote adversarial robustness by robustness in the following.

**Adversarial Training (A.T.).** In contrast to N.T., A.T. aims to improve the robustness of model by incorporating adversarial examples during the training. The goal of A.T. can be expressed as a min-max optimization process:

$$\min_{\boldsymbol{\theta}} \mathbb{E}_{p(\boldsymbol{x}, y)} \left[ \max_{\|\boldsymbol{\gamma}\| \leq \epsilon} \ell_{\mathrm{NT}} \left( f(\boldsymbol{x} + \boldsymbol{\gamma} \mid \boldsymbol{\theta}), y \right) \right], \tag{2}$$

where $\boldsymbol{\gamma}$ is the adversarial perturbation, $\epsilon$ is the adversarial perturbation radius and $\|\cdot\|$ represents the norm operation (i.e., $L_\infty$-norm). Here, the inner layer attempts to find the adversarial examples that causes the classifier to misclassify, while the outer layer focuses on optimizing the model parameters to accurately classify these adversarial examples.

### 2.2 VISUAL REPROGRAMMING

Let $\mathcal{X}_{\mathcal{S}} \subseteq \mathbb{R}^{d_{\mathcal{S}}}$ be the $d_{\mathcal{S}}$-dimensional source domain space and $\mathcal{Y}_{\mathcal{S}} = \{1, \ldots, k_{\mathcal{S}}\}$ be the source domain label space where $k_{\mathcal{S}}$ denotes the number of source domain classes. Assume we have a pre-trained model $f_{\mathcal{S}} : \mathcal{X}_{\mathcal{S}} \to \mathbb{R}^{k_{\mathcal{S}}}$ with parameters $\boldsymbol{\theta}_{\mathrm{S}}$ trained on the source domain dataset $\mathcal{D}_{\mathcal{S}} = \{(\boldsymbol{x}_i^{\mathcal{S}}, y_i^{\mathcal{S}})\}_{i=1}^{N_{\mathcal{S}}}$, where each source domain example $(\boldsymbol{x}_i^{\mathcal{S}}, y_i^{\mathcal{S}}) \in \mathcal{X}_{\mathcal{S}} \times \mathcal{Y}_{\mathcal{S}}$ is assumed to be sampled from an unknown source data distribution with probability density $p(\boldsymbol{x}_{\mathcal{S}}, y_{\mathcal{S}})$. It is noteworthy that $f_{\mathcal{S}}$ is frozen (i.e., $\boldsymbol{\theta}_{\mathrm{S}}$ is non-trainable) in VR, so we will omit $\boldsymbol{\theta}_{\mathrm{S}}$ in following parts. For the target domain of VR, we denote the $d_{\mathcal{T}}$-dimensional target domain space as $\mathcal{X}_{\mathcal{T}} \subseteq \mathbb{R}^{d_{\mathcal{T}}}$, and the target label space as $\mathcal{Y}_{\mathcal{T}} = \{1, \ldots, k_{\mathcal{T}}\}$ where $k_{\mathcal{T}}$ denotes the number of target domain classes. Let $\mathcal{D}_{\mathcal{T}} = \{(\boldsymbol{x}_i^{\mathcal{T}}, y_i^{\mathcal{T}})\}_{i=1}^{N_{\mathcal{T}}}$ be the target domain training set, where each target domain example

$(\boldsymbol{x}_i^{\mathcal{T}}, y_i^{\mathcal{T}}) \subseteq \mathcal{X}_{\mathcal{T}} \times \mathcal{Y}_{\mathcal{T}}$ is assumed to be sampled from an unknown data distribution with probability density $p(\boldsymbol{x}_{\mathcal{T}}, y_{\mathcal{T}})$. Then, the training objective of *natural* visual reprogramming is:

$$\min_{\boldsymbol{\theta}_{\mathrm{in}}, \boldsymbol{\theta}_{\mathrm{out}}} \mathbb{E}_{p(\boldsymbol{x}_{\mathcal{T}}, y_{\mathcal{T}})} \left[ \ell_{\mathrm{NT}}^{\mathcal{T}} ((f_{\mathrm{out}} \circ f_S \circ f_{\mathrm{in}})(\boldsymbol{x}_{\mathcal{T}} \mid \boldsymbol{\theta}_{\mathrm{in}}, \boldsymbol{\theta}_{\mathrm{out}}), y_{\mathcal{T}}) \right], \tag{3}$$

where $f_{\mathrm{in}} : \mathcal{X}_{\mathcal{T}} \to \mathcal{X}_S$ with parameters $\boldsymbol{\theta}_{\mathrm{in}}$ is the visual input transformation, $f_{\mathrm{out}} : \mathbb{R}^{k_S} \to \mathbb{R}^{k_{\mathcal{T}}}$ with parameters $\boldsymbol{\theta}_{\mathrm{out}}$ is the output label mapping, and $\ell_{\mathrm{NT}}^{\mathcal{T}} : \mathbb{R}^{k_{\mathcal{T}}} \times \mathcal{Y}_{\mathcal{T}} \to \mathbb{R}$ is the natural loss function of the target domain. It is worth noting that VR is to reprogram a pre-trained model $f_S$ from a complex source domain to a simpler target domain, so VR assumes (1) the dimension of the target data is not greater than that of the source data (i.e., $d_{\mathcal{T}} \leq d_S$); (2) the number of target class labels is not greater than that of the source class labels (i.e., $k_{\mathcal{T}} \leq k_S$) (Chen, 2022; Cai et al., 2024).

**Visual Input Transformation.** Due to the discrepancies between the input spaces of the target domain and the source domain, data from the target domain cannot be directly used as input for the source domain model. Consequently, it is necessary to transform the data from the target domain. This approach primarily includes *Padding-based* and *Resizing-based* methods. The Padding-based method (Pad) (Tsai et al., 2020; Chen et al., 2023b) adds a learnable parameter $\boldsymbol{\delta}$ around the image from the target domain to ensure the resulting image conforms to the input shape required by the source model. The Resizing-based method, which includes fully-watermarking (Full) (Bahng et al., 2022) and sample-specific multi-channel masks (SMM) (Cai et al., 2024), first resizes the target domain image to the input shape of the source model using interpolation algorithms. Then, the learnable parameter $\boldsymbol{\delta}$ is multiplied by a mask $\boldsymbol{M}$ and added to the resized image. The difference between the two lies in that the mask $\boldsymbol{M}$ in watermarking is a manually set, fixed matrix containing only 0s and 1s, while in SMM, $\boldsymbol{M}$ is generated by a learnable lightweight network that takes the resized image as input and produces an input-dependent mask. Based on the above, we can formalize visual input transformation as:

$$f_{\mathrm{in}}(\boldsymbol{x} \mid \boldsymbol{\theta}_{\mathrm{in}} = (\boldsymbol{\delta}, \boldsymbol{M})) = g(\boldsymbol{x}) + \boldsymbol{\delta} \odot \boldsymbol{M}, \tag{4}$$

where $g(\cdot)$ represents the transformation applied to the target input (e.g., padding or resizing).

**Output Label Mapping.** The output label mapping can be categorized into *non-iterative* and *iterative* methods. Non-iterative methods include one-to-one Random Label Mapping (RandLM) (Elsayed et al., 2018) and one-to-one Frequency-based Label Mapping (FreqLM) (Tsai et al., 2020). RandLM establishes a random mapping between the output space of the source domain and that of the target domain at the beginning of training. FreqLM matches the labels of the source and target domains based on the prediction frequency of the source model on $g(\boldsymbol{x})$ at the beginning of training. Iterative methods include one-to-one Iterative Label Mapping (IterLM) (Chen et al., 2023b) and Fully Label Mapping (FullyLM). IterLM iis an iterative version of the FreqLM method. Specifically, IterLM recalculates the mapping relationship using FreqLM during each round of training. FullyLM employs a fully connected layer and updates its parameters using gradient descent. These output label mapping methods can be uniformly represented as:

$$f_{\mathrm{out}}(\boldsymbol{x} \mid f_{\mathrm{in}}, f_S, \boldsymbol{\theta}_{\mathrm{out}} = \boldsymbol{W}) = \boldsymbol{W}(f_S \circ f_{\mathrm{in}})(\boldsymbol{x} \mid \boldsymbol{\theta}_{\mathrm{in}}) + \boldsymbol{b}. \tag{5}$$

For the one-to-one mapping methods, $\boldsymbol{W} \in \{0, 1\}^{k_{\mathcal{T}} \times k_S}$ is a matrix with at most one element equal to 1 in each row (i.e., $\sum_{j=1}^{k_S} \boldsymbol{W}_{\cdot, j} = 1$), and $\boldsymbol{b}$ is an all-zero vector (i.e., $\boldsymbol{b} = \{0\}^{k_{\mathcal{T}}}$). For FullyLM, $\boldsymbol{W}$ and $\boldsymbol{b}$ are learnable parameters (i.e., $\boldsymbol{W} \in \mathbb{R}^{k_{\mathcal{T}} \times k_S}$ and $\boldsymbol{b} \in \mathbb{R}^{k_{\mathcal{T}}}$).

## 3 VISUAL REPROGRAMMING WITH ADVERSARIAL ROBUSTNESS

We define the label function as $h_f : x \to \arg\max_y [f(x)]_y$, where $f$ is a multi-class score model (e.g., $f_S$ and $f_{\mathrm{out}} \circ f_S \circ f_{\mathrm{in}}$) and $[f(x)]_i$ denotes the $i$-th score output by the model. Then, for a given $h_f$, the expected adversarial risk with respect to the distribution $\mathcal{U}$ is defined as:

$$\mathcal{R}_{\mathcal{U}}^{\mathrm{adv}}(h_f, \epsilon) \triangleq \mathbb{E}_{(\boldsymbol{x}, y) \sim \mathcal{U}} \max_{\|\gamma\| \leq \epsilon} \mathbb{I}[h_f(\boldsymbol{x} + \gamma) \neq y], \tag{6}$$

where $\mathbb{I}[\cdot]$ denotes the indicator function. The adversarial aims for VR is to learn a visual input transformation $f_{\mathrm{in}}$ and output label mapping $f_{\mathrm{out}}$ that tries to minimize adversarial risk $\mathcal{R}_{\mathcal{T}}^{\mathrm{adv}}(f_{\mathrm{out}} \circ f_S \circ f_{\mathrm{in}}, \epsilon)$ in target domain $\mathcal{T}$.

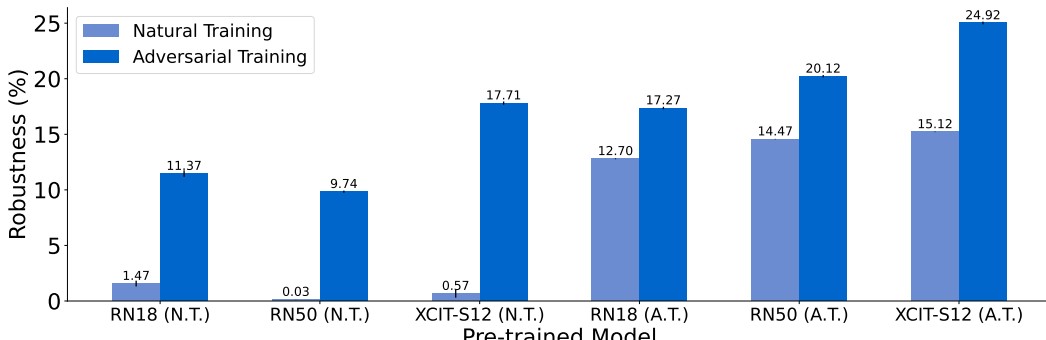

Figure 2: The test performance of three different pre-trained models (e.g., ResNet18, ResNet50, and XCiT-S12, all pre-trained on ImageNet-1K) for reprogramming on the GTSRB dataset. There are two versions for each pre-trained model, including naturally trained models (N.T.) that lack adversarial robustness and adversarially trained models (A.T.) that possess robustness. The visual input transformation is fixed as Pad and the output label mapping is fixed as FullyLM.

Inspired by the effective reuse of well-developed pre-trained models in VR, one may apply Eq. (3) to reprogram an adversarially pre-trained model of the source domain that possesses satisfactory robustness, such as an adversarially trained (A.T.) model, in order to repurpose its robustness. Furthermore, since adversarial training is known to effectively enhance robustness for model, incorporating adversarial examples of target domain into the reprogramming process is anticipated to improve robustness on the target domain, even if the pre-trained model is not inherently robust, such as a naturally trained (N.T.) model. Moreover, simultaneously using a pre-trained model with robustness and incorporating adversarial examples during reprogramming could potentially yield further robustness of the reprogrammed model in the target domain.

To implement adversarial training, we integrate adversarial examples directly into the VR training objective as follows:

$$\min_{\boldsymbol{\theta}} \mathbb{E}_{p(\boldsymbol{x}_{\mathcal{T}}, y_{\mathcal{T}})} \left[ \max_{\|\boldsymbol{\gamma}\| \leq \epsilon} \ell_{\mathrm{NT}}^{\mathcal{T}} \left( (f_{\mathrm{out}} \circ f_{\mathcal{S}} \circ f_{\mathrm{in}})(\boldsymbol{x}_{\mathcal{T}} + \boldsymbol{\gamma} \mid \boldsymbol{\theta}_{\mathrm{in}}, \boldsymbol{\theta}_{\mathrm{out}}), y_{\mathcal{T}} \right) \right], \tag{7}$$

where $\{\boldsymbol{\theta}_{\mathrm{in}}, \boldsymbol{\theta}_{\mathrm{out}}\}$ are the trainable parameters (i.e., visual input transformation and output label mapping) of VR. Furthermore, inspired by TRADES Zhang et al. (2019a), we add a robustness regularization term to the VR training objective, which measures the magnitude of the output differences between the original examples $(\boldsymbol{x}_{\mathcal{T}}, y)$ and the adversarial examples $(\boldsymbol{x}_{\mathcal{T}} + \boldsymbol{\gamma}, y)$. Thus, this robust training objective can be expressed as follows:

$$\min_{\boldsymbol{\theta}} \mathbb{E}_{p(\boldsymbol{x}_{\mathcal{T}}, y_{\mathcal{T}})} [\ell_{\mathrm{NT}}^{\mathcal{T}} ((f_{\mathrm{out}} \circ f_{\mathcal{S}} \circ f_{\mathrm{in}})(\boldsymbol{x}_{\mathcal{T}} \mid \boldsymbol{\theta}_{\mathrm{in}}, \boldsymbol{\theta}_{\mathrm{out}}), y_{\mathcal{T}}) \tag{8}$$
$$+ \lambda \cdot \max_{\|\boldsymbol{\gamma}\| \leq \epsilon} \ell_{\mathrm{KL}}((f_{\mathrm{out}} \circ f_{\mathcal{S}} \circ f_{\mathrm{in}})(\boldsymbol{x}_{\mathcal{T}} \mid \boldsymbol{\theta}_{\mathrm{in}}, \boldsymbol{\theta}_{\mathrm{out}}) \| (f_{\mathrm{out}} \circ f_{\mathcal{S}} \circ f_{\mathrm{in}})(\boldsymbol{x}_{\mathcal{T}} + \boldsymbol{\gamma} \mid \boldsymbol{\theta}_{\mathrm{in}}, \boldsymbol{\theta}_{\mathrm{out}}))]$$

where $\lambda \geq 0$ is a hyperparameter controlling the trade-off between natural loss and robust regularization, and $\ell_{\mathrm{KL}}(P \| Q) = \sum_i P(i) \log \left( \frac{P(i)}{Q(i)} \right)$ denotes the KL divergence which is used to measure the distance between the two probability distributions. In Figure 2, we show the results of reprogramming using adversarial examples generated from the target domain and the results obtained from reprogramming using adversarially pre-trained models.

## 4 THEORETICAL UPPER BOUND

As previously discussed, leveraging adversarially pre-trained models for reprogramming and incorporating adversarial examples from the target domain may both enhance the robustness of reprogrammed models. In this section, we provide the first adversarial risk upper bound for VR, which validates our findings and provides a theoretical guarantee for future research.

## 4.1 Adversarial Margin Loss

Due to the significant role of margin theory in ensuring generalizability, inspired by (Zhang et al., 2019b), we employ margin loss as a substitute for the 0-1 loss. The margin of model $f$ on the example $(\boldsymbol{x}, y)$ is defined as:

$$\rho_f(\boldsymbol{x}, y) \triangleq \frac{1}{2}([f(\boldsymbol{x})]_y - \max_{y' \neq y}[f(\boldsymbol{x})]_{y'}). \tag{9}$$

Then, the corresponding adversarial margin risk can be represented as follows:

$$\mathcal{R}_{\mathcal{U}}^{\mathrm{adv},(\rho)}(f) \triangleq \mathbb{E}_{(\boldsymbol{x},y)\sim\mathcal{U}} \left[ \max_{\|\gamma\|\leq\epsilon} \Phi_\rho \left( \rho_f(\boldsymbol{x} + \gamma, y) \right) \right], \tag{10}$$

where $\Phi_\rho(\cdot)$ is a piecewise function defined as follows:

$$\Phi_\rho(z) \triangleq \begin{cases} 0 & \rho \leq z \\ 1 - z/\rho & 0 \leq z \leq \rho \\ 1 & z \leq 0 \end{cases}. \tag{11}$$

**Lemma 1.** *Given a distribution $\mathcal{U}$ and a socre model $f$, for any $\rho \geq 0$, adversarial risk $\mathcal{R}_{\mathcal{U}}^{\mathrm{adv}}(h_f)$ and adversarial margin risk $\mathcal{R}_{\mathcal{U}}^{\mathrm{adv},(\rho)}(f)$ have the following lemma:*

$$\mathcal{R}_{\mathcal{U}}^{\mathrm{adv},(\rho)}(f) \geq \mathcal{R}_{\mathcal{U}}^{\mathrm{adv}}(h_f).$$

The proof of Lemma 1 is provided in Appendix A.1. Lemma 1 indicates that the adversarial margin risk can serve as an upper bound for the adversarial risk. Based on this relationship, we can derive the adversarial risk upper bound for VR through the margin loss.

## 4.2 Adversarial Margin Disparity Discrepancy

Margin Disparity Discrepancy (MDD) aims to quantify the inconsistency between domains by comparing the marginal performance differences of a classifier across two distinct domains. However, MDD does not effectively quantifies the differences in adversarial marginal performance of the classifier across the domains. Therefore, we propose the definition of Adversarial Margin Disparity Discrepancy of Visual Reprogramming with Adversarial Robustness (AMDD-VRA) within the framework of robustness. Before that, we first define the adversarial margin disparity (AMD) between $f'$ and $f$ on domain $\mathcal{U}$ as:

$$\mathrm{disp}_{\mathcal{U}}^{\mathrm{adv},(\rho)}(f', f) \triangleq \mathbb{E}_{x\sim p(\boldsymbol{x})} \left[ \max_{\|\gamma\|\leq\epsilon} \left( \Phi_\rho(\rho_{f'})(x + \gamma, h_f(x + \gamma)) \right) \right]. \tag{12}$$

AMD quantifies the maximum margin disparity between models $f'$ and $f$ within the $\epsilon$-perturbation range around the instance $\boldsymbol{x}$, meaning the expectation of inconsistent classification between $f'$ and $f$ under $\epsilon$-perturbation in the domain $\mathcal{U}$. Based on AMD, we introduce the definition of AMDD-VRA.

**Definition 1** (AMDD-VRA). *Suppose that $\mathcal{F}_{\mathcal{S}}$ is the hypothesis spaces of the source domain model. For a given pre-trained $f_{\mathcal{S}}$ from source domain, visual input transformation $f_{\mathrm{in}}$ and output label mapping $f_{\mathrm{out}}$, the AMDD-VRA between source domain $\mathcal{S}$ and target domain $\mathcal{T}$ is defined as :*

$$d_{f_{\mathrm{out}}\circ f_{\mathcal{S}}\circ f_{\mathrm{in}}, \mathcal{F}_{\mathcal{S}}}^{\mathrm{adv},(\rho)}(\mathcal{S}, \mathcal{T}) = \sup_{f'_{\mathcal{S}}\in\mathcal{F}_{\mathcal{S}}} \left| \mathrm{disp}_{\mathcal{T}}^{\mathrm{adv},(\rho)}( f_{\mathrm{out}}\circ f'_{\mathcal{S}}\circ f_{\mathrm{in}}, f_{\mathrm{out}}\circ f_{\mathcal{S}}\circ f_{\mathrm{in}}) - \mathrm{disp}_{\mathcal{S}}^{\mathrm{adv},(\rho)}(f'_{\mathcal{S}}, f_{\mathcal{S}}) \right|.$$

From Definition 1, AMDD-VRA measures the adversarial distribution distances between the source domain $\mathcal{S}$ and the target domain $\mathcal{T}$. It can be observed that $f_{\mathrm{in}}$ and $f_{\mathrm{out}}$ influence the value of AMDD-VRA. For instance, when the source domain and target domain are completely identical, if both $f_{\mathrm{in}}$ and $f_{\mathrm{out}}$ are identity mappings, the AMDD-VRA value is 0. However, in the context of VR settings, where the source and target domains are not identical, $f_{\mathrm{in}}$ and $f_{\mathrm{out}}$ can play a role in either narrowing or widening the adversarial discrepancy between the two different domains. Furthermore, based on the definition of AMD, it can be inferred that whether adversarial training is employed for $f_{\mathrm{in}}$ and $f_{\mathrm{out}}$ may impact the first term of AMDD-VRA, thereby affecting the overall value of AMDD-VRA. From the definition of AMDD-VRA and Lemma 1, we can induce the following adversarial risk upper bound for VR:

**Theorem 1.** *Suppose that the $f_{\mathcal{S}}^* = \underset{f_{\mathcal{S}} \in \mathcal{F}_{\mathcal{S}}}{\arg\min} \{\mathcal{R}_{\mathcal{S}}^{\mathrm{adv},(\rho)}(f_{\mathcal{S}})\}$ is the optimal classifier with robustness in source domain. For a given output label mapping $f_{\mathrm{out}}$, pre-trained source domain model $f_{\mathcal{S}}$, and visual input transformation $f_{\mathrm{in}}$, the following adversarial risk upper bound for VR from source domain $\mathcal{S}$ to target domain $\mathcal{T}$ holds:*

$$\mathcal{R}_{\mathcal{T}}^{\mathrm{adv}}(h_{f_{\mathrm{out}} \circ f_{\mathcal{S}} \circ f_{\mathrm{in}}}) \leq \mathcal{R}_{\mathcal{S}}^{\mathrm{adv},(\rho)}(f_{\mathcal{S}}) + d_{f_{\mathrm{out}} \circ f_{\mathcal{S}} \circ f_{\mathrm{in}}, \mathcal{F}_{\mathcal{S}}}^{\mathrm{adv},(\rho)}(\mathcal{S}, \mathcal{T}) + \mathcal{R}_{\mathcal{T}}^{\mathrm{adv},(\rho)}(f_{\mathrm{out}} \circ f_{\mathcal{S}}^* \circ f_{\mathrm{in}}) + \lambda,$$

*where $\lambda = \mathcal{R}_{\mathcal{S}}^{\mathrm{adv},(\rho)}(f_{\mathcal{S}}^*)$ is a constant associated with the source domain.*

The proof of Theorem 1 is provided in Appendix A.2. Theorem 1 reveals that the adversarial risk of VR is influenced by the following factors: 1) the adversarial margin risk of the selected pre-rained model $f_{\mathcal{S}}$ in the source domain; 2) the AMDD-VRA between the source domain and the target domain; 3) the adversarial marginal risk in the target domain for the optimal adversarial trained source domain model $f_{\mathcal{S}}^*$ after reprogramming; 4) the adversarial margin risk of the optimal adversarial trained source domain model $f_{\mathcal{S}}^*$ in the source domain, which can be considered as a constant value $\lambda$. This theory validates our previous findings. In the next section, we will provide more experiments to demonstrate the effectiveness of our strategy.

## 5 EXPERIMENTS

In this section, we empirically demonstrate that by adopting the strategies revealed in our findings, the robustness of reprogrammed models can be enhanced.

### 5.1 EXPERIMENTAL SETUPS

**Pre-trained Source Models and Target Tasks.** We select ImageNet-1K (Russakovsky et al., 2015) (the most commonly used subset of the well-known ImageNet image classification dataset) dataset as the source task. This dataset encompasses 1,000 object classes and contains 1,281,167 training images. The source domain pre-trained models $f_{\mathcal{S}}$ include ResNet-18 (RN18), ResNet-50 (RN50) (He et al., 2016) and XCiT-S12 (El-Nouby et al., 2021), all of which have an input shape of $224 \times 224$. The weights of the naturally pre-trained models (ResNet-18 (N.T.) and ResNet-50 (N.T.)) are obtained from the official PyTorch model repository, while the weights of the adversarially pre-trained models (ResNet-18 (A.T.), ResNet-50 (A.T.) (Salman et al., 2020) and XCiT-S12 (Debenedetti et al., 2023)) are obtained from Robustbench (Croce et al., 2021). For the target datasets, we selected CIFAR-10, CIFAR-100 (Krizhevsky et al., 2009), and GTSRB (Stallkamp et al., 2012), where the target classes are 10, 100, and 43. Detailed information about all datasets can be found in Appendix B. Our goal is to obtain a robust classifier via VR.

**Implementation.** In the implementation of VR, we fix the pre-trained model $f_{\mathcal{S}}$ and select the visual input transformation model $f_{\mathrm{in}}$ from Pad, Full, and SMM, while the output label mapping $f_{\mathrm{out}}$ is chosen from RandLM, IterLM, and FullyLM. Additionally, during the implementation of adversarial training and testing, we generate adversarial examples using 10-step Projected Gradient Descent (Madry et al., 2018) (PGD-10) for training, setting the perturbation radius to $\epsilon = 4/255$ under $L_\infty$-norm, consistent with the settings of adversarially pre-trained models. When using TRADES for adversarial training of VR, we set $\lambda = 6$, which is a commonly used setting in previous adversarial training research. Moreover, we use the AdamW optimizer to train all methods for 60 epochs, and repeat the sampling-and-training process 3 times. The learning rate of the optimizer is searched from set $\{1 \times 10^{-3}, 5 \times 10^{-4}\}$, while the weight decay of the optimizer is searched from set $\{10^{-2}, 10^{-3}\}$.

Details of the experiment, complete results, and some visual results can be found in Appendix B.

### 5.2 EXPERIMENTAL RESULTS

The experiments are conducted on the combinations of pre-trained models and training methods, encompassing four distinct cases: **Strategy 1**: VR with a naturally pre-trained XCiT-S12 and using natural training; **Strategy 2**: VR with a naturally pre-trained XCiT-S12 and using adversarial training; **Strategy 3**: VR with an adversarially pre-trained XCiT-S12 and using natural training; **Strategy 4**: VR with an adversarially pre-trained XCiT-S12 and using adversarial training. Table 1,

Table 1: The performance (mean (%) ± std (%)) comparison of VR with XCiT-S12 on CIFAR10, where PGD-10 is used for adversarial training, and PGD-20 is used for adversarial testing. The results are presented in the format of robustness (clean accuracy), where the std of clean accuracy is omitted. The best robustness is highlighted in bold.

| In / Out | Adversarially Pre-trained Model / VR using Adversarial Training | | | |
|---|---|---|---|---|
| | Strategy 1 (✗ / ✗) | Strategy 2 (✗ / ✔) | Strategy 3 (✔ / ✗) | Strategy 4 (✔ / ✔) |
| Pad / RandLM | 0.06±0.01 (68.49) | 20.97±0.25 (48.84) | 19.55±0.16 (35.70) | **22.84±0.26** (34.81) |
| Pad / FreqLM | 0.02±0.02 (71.38) | 20.98±0.25 (50.65) | 19.71±0.04 (39.86) | **23.56±0.60** (36.69) |
| Pad / IterLM | 0.04±0.02 (72.98) | 19.65±0.67 (45.76) | 20.81±0.06 (40.93) | **22.79±1.20** (35.90) |
| Pad / FullyLM | 0.13±0.02 (80.72) | 25.37±0.47 (53.71) | 31.84±0.01 (67.72) | **43.59±0.07** (64.04) |
| Full / RandLM | 0.00±0.00 (70.13) | 19.58±0.81 (40.82) | 4.00±0.28 (49.60) | **21.92±4.01** (35.19) |
| Full / FreqLM | 0.10±0.08 (88.43) | 24.36±1.77 (49.82) | 20.87±0.32 (66.64) | **28.91±0.45** (55.64) |
| Full / IterLM | 0.12±0.08 (87.98) | 19.93±2.25 (39.66) | 23.40±0.72 (71.65) | **25.62±1.47** (44.29) |
| Full / FullyLM | 12.09±0.74 (93.84) | 44.39±0.26 (74.48) | 38.52±0.01 (91.96) | **57.09±0.08** (89.62) |

Table 2: The performance (mean (%) ± std (%)) comparison of VR with XCiT-S12 on CIFAR100, where PGD-10 is used for adversarial training, and PGD-20 is used for adversarial testing. The results are presented in the format of robustness (clean accuracy), where the std of clean accuracy is omitted. The best robustness is highlighted in bold.

| In / Out | Adversarially Pre-trained Model / VR using Adversarial Training | | | |
|---|---|---|---|---|
| | Strategy 1 (✗ / ✗) | Strategy 2 (✗ / ✔) | Strategy 3 (✔ / ✗) | Strategy 4 (✔ / ✔) |
| Pad / RandLM | 0.91±0.33 (22.27) | 1.10±0.14 (1.73) | **1.13±0.09** (2.28) | 0.79±0.01 (1.27) |
| Pad / FreqLM | 0.10±0.06 (29.10) | **4.54±0.19** (10.88) | 3.06±0.16 (6.35) | 2.52±0.12 (4.05) |
| Pad / IterLM | 0.14±0.01 (38.54) | 0.75±0.02 (1.79) | 4.61±0.44 (11.02) | **5.26±0.36** (8.90) |
| Pad / FullyLM | 0.46±0.02 (58.32) | 12.07±0.35 (36.35) | 18.40±0.12 (44.29) | **25.93±0.08** (43.59) |
| Full / RandLM | 0.00±0.00 (19.15) | 1.01±0.01 (1.15) | 0.74±0.18 (6.32) | **1.19±0.20** (1.89) |
| Full / FreqLM | 0.04±0.01 (49.84) | 7.34±1.90 (17.32) | 9.66±0.13 (32.75) | **10.62±0.28** (30.49) |
| Full / IterLM | 0.18±0.03 (57.63) | 2.74±0.74 (4.64) | 11.29±0.17 (38.07) | **11.93±0.30** (33.21) |
| Full / FullyLM | 0.78±0.13 (77.28) | 24.55±0.20 (52.91) | 22.02±0.11 (74.06) | **37.99±0.21** (72.88) |

Table 2, and Table 3 present the performance comparison results of VR with pre-trained XCiT-S12 on CIFAR-10, CIFAR-100, and GTSRB, respectively.

**Performance of VR with Naturally and Adversarially Pre-trained XCiT-S12.** The results indicate that in Strategy 1, despite achieving very high clean accuracy, the robustness of various VR methods is nearly zero. For instance, when using naturally pre-trained XCiT-S12 in Strategy 1, "Pad / FullyLM" achieves a clean accuracy of 65.30% on GTSRB (Table 3), yet its robustness is 0.57%; similarly, "Full / IterLM" achieves a clean accuracy of 57.63% on CIFAR-100 (Table 2), but also exhibits an adversarial robustness of 0.18%. Although in Strategy 1, "Full / FullyLM" achieves a clean accuracy of 93.84% and robustness of 12.09% on CIFAR-10 (Table 1), the rest of the VR methods all achieve almost zero robustness. This demonstrates the extreme vulnerability of VR with naturally pre-trained models to adversarial attacks. Furthermore, when using adversarilly pre-trained XCiT-S12 in Strategy 3, although there is a general decrease in clean accuracy compared with Strategy 1, most VR methods displayed some level of robustness. For example, on CIFAR-10, "Pad / FullyLM" achieves a clean accuracy of 67.72% with an adversarial robustness of 31.84%; "Full / IterLM" shows a clean accuracy of 71.65% with an adversarial robustness of 23.40%. This suggests that existing VR methods can leverage the robustness of pre-trained models to some extent.

Similarly, comparing the experimental results of Strategy 2 and Strategy 4, it is evident that the robustness of using an adversarially pre-trained XCiT-S12 is mostly higher than using a naturally

pre-trained XCiT-S12. For instance, the robustness of "Pad / RandLM" on CIFAR-10 is 20.97% in Strategy 2, which is lower than 22.84% in Strategy 4. Consistent findings are also evident across other datasets and various VR methods.

These phenomenon is consistent with our theoretical analysis: the adversarial risk of VR in target domain is upper-bounded by the adversarial margin risk of the pre-rained model in the source domain. Specifically, compared with naturally pre-trained models, adversarially pre-trained models generally exhibit better robustness (see Appendix B), resulting in lower adversarial margin risk, which may help reduce the upper bound of adversarial risk in the target domain. Therefore, models reprogrammed using adversarially pre-trained models will have higher robustness compared whit those reprogrammed using naturally pre-trained models.

**Performance of VR with Naturally and Adversarially Training.** When using adversarial training in Strategy 2, there is also a decrease in clean accuracy compared with Strategy 1 (using natural training). Similarly, the robustness of VR in Strategy 2 show an improvement compared with Strategy 1. In fact, when we incorporate adversarial samples from target domain into the reprogramming process by adversarial training, we are directly optimizing the adversarial risk of VR, so the observed empirical phenomena are consistent with previous adversarial training research. For example, in the "Pad / FullyLM" on Table 1, the Strategy 2 achieved a robustness improvement of 25.24% compared with Strategy 1; in the "Full / FullyLM", the Strategy 2 achieved a robustness improvement of 32.30% compared with Strategy 1. Similar results can be observed in all experiments. This is consistent with the theoretical analysis that incorporating adversarial samples from the target domain task into the reprogramming process of VR through adversarial training may reduce the ADMM-VAR, which can enhance the robustness of the reprogrammed models.

Similarly, by comparing the experimental results of Strategy 3 and Strategy 4, it is evident that the robustness achieved through adversarial training during the reprogramming process is higher than that achieved through natural training. It can be observed that the robustness improvements achieved through adversarial training during VR are relatively limited compared with those obtained using adversarially pre-trained models. For instance, as shown in Table 1, the improvement of Strategy 4 over Strategy 3 is generally smaller than that of Strategy 3 over Strategy 1. The limited robustness improvements from adversarial training may be due to the finite number of trainable parameters in VR. As illustrated in Tables 1 to 3 and Figure 3, the robustness enhancement through adversarial training becomes more pronounced when VR methods with larger parameter spaces (e.g., SMM and FullyLM) are employed. Therefore, adversarial training remains a crucial technique for improving robustness. Developing adversarial training methods tailored to the VR framework and designing more effective $f_{in}$ and $f_{out}$ represent important challenges for enhancing the robustness of VR.

**Performance of VR under Different Adversarial Attack Intensities.** We also evaluate the robustness of the VR under different adversarial attack intensities. Specifically, on the CIFAR-10 and GT-SRB, we employ various VR methods using adversarially pre-trained ResNet-18 with TRADES as adversarial training. During the testing phase, we modify two key parameters of the PGD during the testing phase: setting the adversarial perturbation radius from $\{1/255, 2/255, 4/255, 6/255, 8/255\}$ while fixing the iteration number at 20, and setting the number of iterations from $\{1, 10, 20, 50, 100\}$ while fixing the adversarial perturbation radius at $4/255$. The results are displayed in Figure 3. As illustrated in Figure 3, although increasing either the adversarial perturbation radius or the number of PGD iterations leads to a reduction in the adversarial accuracy of the various VR methods, their relative robustness rankings largely remain unchanged. These results indicate that VR with an adversarially pre-trained model and using adversarial training remains robust when facing adversarial attacks of varying intensity.

Additionally, it can be clearly observed from Figure 3 that, when $f_{out}$ is fixed, the robustness gap caused by different $f_{out}$ is significantly smaller on CIFAR-10 compared with GTSRB. This may be because both ImageNet and CIFAR-10 involve natural image classification, whereas GTSRB focuses on traffic signs, which differ from ImageNet in terms of style and context. Moreover, AMDD-VRA is influenced not only by the domain gap but also by the choice of $f_{in}$ and $f_{out}$. Consequently, when the domain gap is large, a better $f_{in}$ can more effectively reduce these differences.

**Interesting Phenomenon about Clean Accuracy.** Moreover, we observed an interesting phenomenon: when performing adversarial training on a naturally pre-trained model by incorporating adversarial samples during reprogramming (i.e., Strategy 2), it achieves a certain level of robustness

Table 3: The performance (mean (%) ± std (%)) comparison of VR with XCiT-S12 on GTSRB, where PGD-10 is used for adversarial training, and PGD-20 is used for adversarial testing. The results are presented in the format of robustness (clean accuracy), where the std of clean accuracy is omitted. The best robustness is highlighted in bold.

| In / Out | Adversarially Pre-trained Model / VR using Adversarial Training | | | |
| --- | --- | --- | --- | --- |
| | Strategy 1 (✘ / ✘) | Strategy 2 (✘ / ✔) | Strategy 3 (✔ / ✘) | Strategy 4 (✔ / ✔) |
| Pad / RandLM | 1.73±0.35 (44.11) | **12.07±1.01** (31.88) | 5.82±0.83 (10.22) | 8.19±0.34 (12.96) |
| Pad / FreqLM | 0.24±0.01 (47.11) | **14.22±0.56** (31.26) | 8.43±0.10 (14.62) | 9.19±1.40 (15.94) |
| Pad / IterLM | 0.16±0.17 (47.54) | 4.72±1.20 (9.15) | 8.68±0.27 (15.83) | **9.79±0.77** (12.04) |
| Pad / FullyLM | 0.57±0.37 (65.30) | 17.71±0.04 (44.09) | 15.12±0.02 (43.36) | **24.92±0.12** (42.76) |
| Full / RandLM | 0.00±0.00 (74.41) | 23.21±0.75 (44.66) | 5.87±1.02 (40.64) | **23.98±0.57** (37.37) |
| Full / FreqLM | 0.00±0.00 (77.14) | 11.60±1.01 (51.63) | 5.70±0.11 (37.67) | **27.04±0.71** (41.30) |
| Full / IterLM | 0.00±0.00 (76.03) | 13.13±2.06 (57.33) | 5.15±0.45 (41.80) | **29.58±1.63** (48.34) |
| Full / FullyLM | 1.74±0.21 (88.70) | 42.66±0.42 (87.89) | 20.85±0.25 (80.62) | **50.37±0.43** (79.88) |

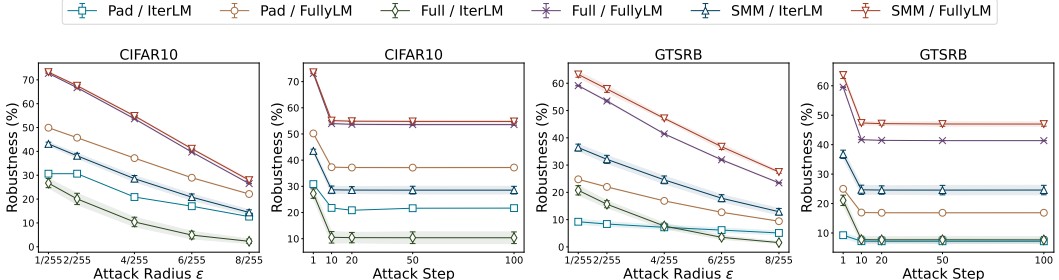

Figure 3: The performance (robustness accuracy) of using TRADES for adversarial reprogramming an adversarially pre-trained ResNet-18 model on the CIFAR-10 and GTSRB datasets as the intensity of adversarial attacks increases.

but suffers a significant drop in clean accuracy. In contrast, when performing adversarial training on an adversarially pre-trained model by incorporating adversarial samples during reprogramming (i.e., Strategy 4), it maintains a good level of robustness while also preserving a certain clean accuracy. For example, in Table 2, the clean accuracy of "Pad / FullyLM" in Strategy 2 drops to 36.35%, which is lower than 43.59% of "Pad / FullyLM" in Strategy 4; the clean accuracy of "Full / FullyLM" in Strategy 4 is 72.88% higher than that of "Full / FullyLM" in Strategy 2. This phenomenon can be observed in most of the experimental results, indicating that the effectiveness (in terms of robustness and clean accuracy) of adversarial training on naturally pre-trained models is inferior to that of adversarial training on adversarially pre-trained models.

## 6 CONCLUSION

In this paper, we for the first time investigate the adversarial robustness of VR, which is unexplored in previously visual reprogramming (VR) methods. Our empirical results reveal that employing naturally pre-trained models and natural training are particularly vulnerable to adversarial attacks, while using pre-trained models with adversarial robustness and adversarial training can both significantly enhance the robustness of reprogrammed models. Additionally, we present a theoretically guaranteed adversarial risk upper bound for VR, validating our empirical findings and laying a theoretical foundation for future research. Extensive experiments demonstrate the effectiveness of the strategies revealed in our empirical findings in enhancing the robustness of VR. Given the substantial potential of VR in diverse tasks and modalities, our strategies hold potential applications in security-critical multimodal large language models. However, the robustness endowed by these strategies remains to be enhanced. In future research, we will explore improved methods for VR based on our results.

ACKNOWLEDGMENTS

This work was supported by the National Key R&D Program of China under Grant 2022YFB3103500. Feng Liu is supported by the Australian Research Council (ARC) with grant number LP240100101, DE240101089 and DP230101540 and the NSF&CSIRO Responsible AI program with grant number 2303037.

**Reproducibility Statement.** For the datasets used in our paper, the detailed descriptions can be found in Appendix B. For the experimental process in our paper, the complete implementation details can be found in Appendix B. For the lemmas and theorems in our paper, the complete proofs can be found in Appendix A. The code for the experiments can be found in supplementary material. These can help reproduce the theoretical and experimental results.

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

## A    PROOFS OF THE ADVERSARIAL RISK UPPER BOUND

### A.1    PROOF OF LEMMA 1

*Proof.* We can discuss the following two scenarios based on whether the classification is correct:

1) If $h_f(\boldsymbol{x}) \neq y$: since the goal of adversarial attack is to induce incorrect predictions by the model, we are able to identify a $\gamma$ ($\|\gamma\| \leq \epsilon$) such that $\mathbb{I}[h_f(\boldsymbol{x} + \gamma) \neq y] = 1$ and $\rho_f(\boldsymbol{x} + \boldsymbol{\gamma}, y) \leq 0$. Therefore, we have $\max_{\|\gamma\| \leq \epsilon} \mathbb{I}[h_f(\boldsymbol{x} + \gamma) \neq y] = \max_{\|\gamma\| \leq \epsilon}(\Phi_\rho \circ \rho_f)(x + \gamma, y) = 1$.

2) If $h_f(\boldsymbol{x}) = y$: if for any $\boldsymbol{\gamma}$ ($\|\gamma\| \leq \epsilon$) such that $\mathbb{I}[h_f(\boldsymbol{x} + \gamma) \neq y] = 0$, then for any $\rho \geq 0$ we have $\max_{\|\gamma\| \leq \epsilon} \mathbb{I}[h_f(\boldsymbol{x} + \gamma) \neq y] = 0 \leq \max_{\|\gamma\| \leq \epsilon}(\Phi_\rho \circ \rho_f) \leq 1$; if there is a $\boldsymbol{\gamma}$ ($\|\gamma\| \leq \epsilon$) such that $\mathbb{I}[h_f(\boldsymbol{x} + \gamma) \neq y] = 1$, the result is equal to scenario 1).

Combining 1) and 2), we have $\mathcal{R}_{\mathcal{U}}^{\mathrm{adv},(\rho)}(f) \geq \mathcal{R}_{\mathcal{U}}^{\mathrm{adv}}(h_f)$ for any $\rho \geq 0$. $\qquad\square$

### A.2    PROOF OF THEOREM 1

**Lemma 2.** *For any distribution $U = p(\boldsymbol{x}, y)$ and any sorce function $f$, we have:*

$$\mathrm{disp}_U^{\mathrm{adv},(\rho)}(f', f) \leq \mathcal{R}_U^{\mathrm{adv},(\rho)}(f') + \mathcal{R}_U^{\mathrm{adv},(\rho)}(f) \tag{13}$$

*Proof.* For any $(\boldsymbol{x}, y) \sim \mathcal{U}$, if there is a $\gamma$ ($\|\gamma\| \leq \epsilon$) such that $h_{f'}(\boldsymbol{x} + \boldsymbol{\gamma}) \neq y$ or $h_f(\boldsymbol{x} + \boldsymbol{\gamma}) \neq y$, then we have $\max_{\|\gamma\| \leq \epsilon}(\Phi_\rho \circ \rho_{f'})(x + \gamma, y) = 1$ or $\max_{\|\gamma\| \leq \epsilon}(\Phi_\rho \circ \rho_f)(x + \gamma, y) = 1$, while $\max_{\|\gamma\| \leq \epsilon}(\Phi_\rho \circ \rho_{f'})(x + \gamma, h_f(x + \gamma)) \leq 1$. If for any $\gamma$ ($\|\gamma\| \leq \epsilon$) such that $h_{f'}(\boldsymbol{x} + \boldsymbol{\gamma}) = h_f(\boldsymbol{x} + \boldsymbol{\gamma}) = y$, we have:

$$\max_{\|\gamma\| \leq \epsilon}(\Phi_\rho \circ \rho_{f'})(x + \gamma, h_f(x + \gamma))$$
$$= \max_{\|\gamma\| \leq \epsilon}(\Phi_\rho \circ \rho_{f'})(x + \gamma, y)$$
$$\leq \max_{\|\gamma\| \leq \epsilon}(\Phi_\rho \circ \rho_{f'})(x + \gamma, y) + \max_{\|\gamma\| \leq \epsilon}(\Phi_\rho \circ \rho_f)(x + \gamma, y).$$

Combining the aforementioned two scenarios and taking expectations, we can obtain the result. $\quad\square$

By combining Lemma 1, we can proof Theorem 1:

*Proof.*

$$\mathcal{R}_{\mathcal{T}}^{\mathrm{adv}}(h_{f_{\mathrm{out}} \circ f_\mathcal{S} \circ f_{\mathrm{in}}}) = \mathbb{E}_{\mathcal{T}} \max_{\|\gamma\| \leq \epsilon} \mathbb{I}[h_{f_{\mathrm{out}} \circ f_\mathcal{S} \circ f_{\mathrm{in}}}(x + \gamma) \neq y]$$

$$\overset{(a)}{\leq} \mathbb{E}_{\mathcal{T}} \max_{\|\gamma\| \leq \epsilon} \left\{ \mathbb{I}[h_{f_{\mathrm{out}} \circ f_\mathcal{S} \circ f_{\mathrm{in}}}(x + \gamma) \neq h_{f_{\mathrm{out}} \circ f_\mathcal{S}^* \circ f_{\mathrm{in}}}(x + \gamma)] + \mathbb{I}[h_{f_{\mathrm{out}} \circ f_\mathcal{S}^* \circ f_{\mathrm{in}}}(x + \gamma) \neq y] \right\}$$

$$\overset{(b)}{\leq} \mathbb{E}_{\mathcal{T}} \left\{ \max_{\|\gamma\| \leq \epsilon} \mathbb{I}[h_{f_{\mathrm{out}} \circ f_\mathcal{S} \circ f_{\mathrm{in}}}(x + \gamma) \neq h_{f_{\mathrm{out}} \circ f_\mathcal{S}^* \circ f_{\mathrm{in}}}(x + \gamma)] + \max_{\|\gamma\| \leq \epsilon} \mathbb{I}[h_{f_{\mathrm{out}} \circ f_\mathcal{S}^* \circ f_{\mathrm{in}}}(x + \gamma) \neq y] \right\}$$

$$= \mathbb{E}_{\mathcal{T}} \max_{\|\gamma\| \leq \epsilon} \mathbb{I}[h_{f_{\mathrm{out}} \circ f_\mathcal{S} \circ f_{\mathrm{in}}}(x + \gamma) \neq h_{f_{\mathrm{out}} \circ f_\mathcal{S}^* \circ f_{\mathrm{in}}}(x + \gamma)] + \mathbb{E}_{\mathcal{T}} \max_{\|\gamma\| \leq \epsilon} \mathbb{I}[h_{f_{\mathrm{out}} \circ f_\mathcal{S}^* \circ f_{\mathrm{in}}}(x + \gamma) \neq y]$$

$$\overset{(c)}{\leq} \mathrm{disp}_{\mathcal{T}}^{\mathrm{adv},(\rho)}(f_{\mathrm{out}} \circ f_\mathcal{S}^* \circ f_{\mathrm{in}}, f_{\mathrm{out}} \circ f_\mathcal{S} \circ f_{\mathrm{in}}) + \mathcal{R}_{\mathcal{T}}^{\mathrm{adv},(\rho)}(f_{\mathrm{out}} \circ f_\mathcal{S}^* \circ f_{\mathrm{in}})$$

$$= \mathrm{disp}_{f_{\mathrm{out}} \circ \mathcal{T}}^{\mathrm{adv},(\rho)}(f_{\mathrm{out}} \circ f_\mathcal{S}^* \circ f_{\mathrm{in}}, f_{\mathrm{out}} \circ f_\mathcal{S} \circ f_{\mathrm{in}})$$
$$\qquad\qquad + \mathcal{R}_{\mathcal{T}}^{\mathrm{adv},(\rho)}(f_{\mathrm{out}} \circ f_\mathcal{S}^* \circ f_{\mathrm{in}}) + \mathcal{R}_{\mathcal{S}}^{\mathrm{adv},(\rho)}(f_\mathcal{S}) - \mathcal{R}_{\mathcal{S}}^{\mathrm{adv},(\rho)}(f_\mathcal{S})$$

$$\overset{(d)}{\leq} \mathrm{disp}_{\mathcal{T}}^{\mathrm{adv},(\rho)}(f_{\mathrm{out}} \circ f_\mathcal{S}^* \circ f_{\mathrm{in}}, f_{\mathrm{out}} \circ f_\mathcal{S} \circ f_{\mathrm{in}}) + \mathcal{R}_{\mathcal{T}}^{\mathrm{adv},(\rho)}(f_{\mathrm{out}} \circ f_\mathcal{S}^* \circ f_{\mathrm{in}})$$
$$\qquad\qquad + \mathcal{R}_{\mathcal{S}}^{\mathrm{adv},(\rho)}(f_\mathcal{S}) + \mathcal{R}_{\mathcal{S}}^{\mathrm{adv},(\rho)}(f_\mathcal{S}^*) - \mathrm{disp}_{\mathcal{S}}^{\mathrm{adv},(\rho)}(f_\mathcal{S}^*, f_\mathcal{S}) \text{ (Lemma 2)}$$

$$\overset{(e)}{\leq} \mathcal{R}_{\mathcal{S}}^{\mathrm{adv},(\rho)}(f_\mathcal{S}) + d_{f_{\mathrm{out}} \circ f_\mathcal{S} \circ f_{\mathrm{in}}, \mathcal{F}_\mathcal{S}}^{\mathrm{adv},(\rho)}(\mathcal{S}, \mathcal{T}) + \mathcal{R}_{\mathcal{T}}^{\mathrm{adv},(\rho)}(f_{\mathrm{out}} \circ f_\mathcal{S}^* \circ f_{\mathrm{in}}) + \lambda$$

where $f_{\mathcal{S}}^* = \underset{f_{\mathcal{S}} \in \mathcal{F}_{\mathcal{S}}}{\arg\min}\{\mathcal{R}_{\mathcal{S}}^{\mathrm{adv},(\rho)}(f_{\mathcal{S}})\}$ and $\lambda = \mathcal{R}_{\mathcal{S}}^{\mathrm{adv},(\rho)}(f_{\mathcal{S}}^*)$ is a constant associated with the source domain. The proof is completed. $\qquad\square$

## B  ADDITIONAL INFORMATION OF EXPERIMENTS

### B.1  ADVERSARIAL EXAMPLE GENERATION

In adversarial training (e.g., PGD-10 or TRADES) and validating model robustness, we employ the iterative projected gradient descent method to generate adversarial examples. Assuming the total number of iterations is $N$, the adversarial example at step $t + 1 \leq N$ is represented as follows:

$$\boldsymbol{x}_{\mathrm{adv}}^{t+1} = \Pi\left(\boldsymbol{x}_{\mathrm{adv}}^t + \alpha \cdot \mathrm{sign}(\nabla_{\boldsymbol{x}_{\mathrm{adv}}^t}\ell_{\mathrm{NT}}\left(f(\boldsymbol{x}_{\mathrm{adv}}^t \mid \boldsymbol{\theta}), y)\right)\right), \tag{14}$$

where the $\Pi$ denotes the porjection, $\alpha$ denotes the step size, and $\mathrm{sign}(\cdot)$ evaluates to $-1, 0$, or $+1$ for each component of the gradient, depending on whether the component is negative, zero, or positive, respectively. Assuming that $\boldsymbol{x}$ represents an image, the projection of the value in $i$-th row and $j$-th column of $c$-th channel is as follows:

$$\boldsymbol{x}_{\mathrm{adv}}^{t+1}[i, j, c] = \mathrm{clip}(\boldsymbol{x}_{\mathrm{adv}}^t[i, j, c], \boldsymbol{x}[i, j, c] - \epsilon, \boldsymbol{x}[i, j, c] + \epsilon), \tag{15}$$

where $\epsilon$ denotes the perturbation radius and clip denotes the operation that restricts the value within a specified range. Similarly, in TRADES, adversarial examples are obtained by maximizing the KL divergence as follows:

$$\boldsymbol{x}_{\mathrm{adv}}^{t+1} = \Pi\left(\boldsymbol{x}_{\mathrm{adv}}^t + \alpha \cdot \mathrm{sign}(\nabla_{\boldsymbol{x}_{\mathrm{adv}}^t}\ell_{\mathrm{KL}}\left(f(\boldsymbol{x}_{\mathrm{adv}}^t \mid \boldsymbol{\theta}), f(\boldsymbol{x} \mid \boldsymbol{\theta}))\right)\right). \tag{16}$$

The detailed process is presented in Algorithm 1.

---

**Algorithm 1** Adversarial Example Generation using Iterative Projected Gradient Descent

---

**Input:** Original input $\boldsymbol{x}$, True label $y$, Model parameters $\boldsymbol{\theta}$, Loss function $\ell_{\mathrm{NT}}$, Step size $\alpha$, Perturbation radius $\epsilon$, Total iterations $N$
**Output:** Adversarial example $\boldsymbol{x}_{\mathrm{adv}}$
1: Initialize $\boldsymbol{x}_{\mathrm{adv}}^0 \leftarrow \boldsymbol{x}$
2: **for** $t = 0$ to $N - 1$ **do**
3:     Compute gradient: $\nabla_{\boldsymbol{x}}\ell_{\mathrm{NT}}\left(f(\boldsymbol{x}_{\mathrm{adv}}^t \mid \boldsymbol{\theta}), y\right)$
4:     Update adversarial example:
5:       $\boldsymbol{x}_{\mathrm{adv}}^{t+1} \leftarrow \boldsymbol{x}_{\mathrm{adv}}^t + \alpha \cdot \mathrm{sign}\left(\nabla_{\boldsymbol{x}_{\mathrm{adv}}^t}\ell_{\mathrm{NT}}\left(f(\boldsymbol{x}_{\mathrm{adv}}^t \mid \boldsymbol{\theta}), y\right)\right)$
6:     Project onto $\epsilon$-ball around $\boldsymbol{x}$:
7:     **for** each pixel $(i, j, c)$ **do**
8:       $\boldsymbol{x}_{\mathrm{adv}}^{t+1}[i, j, c] \leftarrow \mathrm{clip}\left(\boldsymbol{x}_{\mathrm{adv}}^{t+1}[i, j, c], \boldsymbol{x}[i, j, c] - \epsilon, \boldsymbol{x}[i, j, c] + \epsilon\right)$
9:     **end for**
10:     $\boldsymbol{x}_{\mathrm{adv}} \leftarrow \boldsymbol{x}_{\mathrm{adv}}^{t+1}$
11: **end for**
12: **Return** $\boldsymbol{x}_{\mathrm{adv}}$

---

Table 4: Performance of different pre-trained models on source domain (ImageNet-1K). Robustness represents the accuracy of PGD-20 with $\epsilon = 4/255$ and $\epsilon = 8/255$ under $L_\infty$-norm.

| Performance | Adversarially Pre-trained Model | | | | | |
|---|---|---|---|---|---|---|
| | RN18 (✘) | RN50 (✘) | XCiT-S12 (✘) | RN18 (✔) | RN50 (✔) | XCiT-S12 (✔) |
| Clean Acc. | 69.75 | 76.13 | 82.00 | 48.42 | 60.49 | 70.60 |
| Robustness ($\epsilon = 4/255$) | 0.00 | 0.00 | 0.00 | 26.07 | 35.45 | 39.86 |
| Robustness ($\epsilon = 8/255$) | 0.00 | 0.00 | 0.00 | 9.88 | 14.26 | 14.30 |

Table 5: The performance (mean (%) $\pm$ std (%)) comparison of VR with ResNet-18 on CIFAR10, where PGD-10 is used for adversarial training, and PGD-20 is used for adversarial testing. The results are presented in the format of robustness (clean accuracy), where the std of clean accuracy is omitted. The best robustness is highlighted in bold.

| In / Out | Adversarially Pre-trained Model / VR using Adversarial Training | | | |
|---|---|---|---|---|
| | Strategy 1 (✘ / ✘) | Strategy 2 (✘ / ✔) | Strategy 3 (✔ / ✘) | Strategy 4 (✔ / ✔) |
| Pad / RandLM | 0.12±0.02 (56.72) | 8.24±0.71 (25.86) | 17.37±1.17 (26.59) | **17.66±1.27** (24.98) |
| Pad / FreqLM | 0.12±0.04 (58.41) | 6.62±1.66 (26.22) | 18.69±0.83 (29.69) | **20.92±0.51** (29.18) |
| Pad / IterLM | 0.14±0.05 (58.69) | 4.70±2.43 (26.62) | 20.01±0.88 (32.49) | **20.85±0.58** (30.83) |
| Pad / FullyLM | 0.11±0.01 (71.62) | 17.59±0.25 (40.16) | 32.15±0.16 (56.70) | **37.72±0.12** (53.61) |
| Full / RandLM | 0.00±0.00 (61.78) | 3.04±0.99 (29.68) | 4.51±1.21 (41.13) | **11.38±0.67** (28.16) |
| Full / FreqLM | 0.00±0.00 (70.15) | 3.58±0.91 (32.25) | 13.85±0.40 (56.05) | **18.64±0.07** (45.69) |
| Full / IterLM | 0.00±0.00 (69.22) | 4.45±0.85 (30.25) | **17.77±4.22** (58.01) | 13.78±2.37 (39.91) |
| Full / FullyLM | 0.40±0.03 (86.18) | 24.17±0.70 (44.10) | 35.20±0.21 (87.15) | **54.15±0.03** (82.92) |

Table 6: The performance (mean (%) + std (%)) comparison of VR with ResNet-18 on CIFAR100, where PGD-10 is used for adversarial training, and PGD-20 is used for adversarial testing. The results are presented in the format of robustness accuracy (clean accuracy), where the std of clean accuracy is omitted. The best robustness is highlighted in bold.

| In / Out | Adversarially Pre-trained Model / VR using Adversarial Training | | | |
|---|---|---|---|---|
| | Strategy 1 (✘ / ✘) | Strategy 2 (✘ / ✔) | Strategy 3 (✔ / ✘) | Strategy 4 (✔ / ✔) |
| Pad / RandLM | 0.06±0.04 (8.42) | 0.46±0.50 (1.05) | **0.79±0.24** (1.42) | 0.78±0.43 (1.16) |
| Pad / FreqLM | 0.06±0.01 (13.46) | 0.28±0.21 (1.35) | **2.11±0.12** (3.69) | 1.92±0.15 (2.74) |
| Pad / IterLM | 0.09±0.02 (20.41) | 0.64±0.28 (1.45) | **5.42±0.28** (8.96) | 5.04±0.27 (7.58) |
| Pad / FullyLM | 0.29±0.04 (46.81) | 6.36±0.09 (23.87) | 18.37±0.19 (34.24) | **21.17±0.02** (32.61) |
| Full / RandLM | 0.00±0.00 (12.19) | 0.18±0.29 (0.89) | **0.82±0.07** (4.40) | 0.64±0.26 (1.27) |
| Full / FreqLM | 0.00±0.00 (29.79) | 0.07±0.03 (2.06) | 7.91±0.16 (25.50) | **8.50±0.04** (23.42) |
| Full / IterLM | 0.00±0.00 (36.87) | 0.50±0.15 (3.54) | 10.25±0.18 (31.77) | **10.88±0.12** (28.84) |
| Full / FullyLM | 0.07±0.02 (68.33) | 11.87±0.07 (30.31) | 21.91±0.12 (67.75) | **36.57±0.12** (65.55) |

## B.2 EVALUATION METRICS

In the testing phase, we employ the classification accuracy on adversarial test examples as the robustness, where the adversarial test examples are generated from clean examples in the test dataset using 20-step Projected Gradient Descent (PGD-20) with a perturbation radius of $\epsilon = 4/255$ under the $L_\infty$-norm. Let the classification accuracy of the clean test examples be the clean accuracy. In addition, we provide the results for different perturbation radii and different iteration steps in Figure 3.

## B.3 IMAGENET-1K

This is the most commonly used subset of the famous image classification dataset ImageNet (Russakovsky et al., 2015). The dataset covers 1,000 object classes and includes 1,281,167 training images, 50,000 validation images, and 100,000 test images. We set this task as our source domain because the dataset has a large amount of training examples and a wide variety of classification categories, which enables the pre-training of excellent models. Based on this, we use three pre-trained models on this dataset for reprogramming: ResNet-18 (RN18), ResNet-50 (RN50) (He et al., 2016) and XCiT-S12 (El-Nouby et al., 2021). Their input layers are all $224 \times 224$. The weights of the naturally pre-trained models (ResNet-18 (N.T.) and ResNet-50 (N.T.)) are obtained from the official

Table 7: The performance (mean (%) + std (%)) comparison of reprogrammed ResNet-18 on GT-SRB, where PGD-10 is used for adversarial training, and PGD-20 is used for adversarial testing. The results are presented in the format of robustness accuracy (clean accuracy), where the std of clean accuracy is omitted. The best robustness is highlighted in bold.

| In / Out | Adversarially Pre-trained Model / VR using Adversarial Training | | | |
|---|---|---|---|---|
| | Strategy 1 (✗ / ✗) | Strategy 2 (✗ / ✔) | Strategy 3 (✔ / ✗) | Strategy 4 (✔ / ✔) |
| Pad / RandLM | 0.94±0.30 (30.49) | 0.82±0.62 (11.19) | **5.12±0.20** (7.01) | 4.09±1.88 (5.87) |
| Pad / FreqLM | 0.22±0.13 (35.02) | 1.96±0.24 (15.07) | **6.23±0.53** (10.48) | 5.91±0.20 (9.04) |
| Pad / IterLM | 0.31±0.21 (37.70) | 4.95±0.31 (6.15) | 7.17±0.40 (10.30) | **8.44±0.71** (11.33) |
| Pad / FullyLM | 1.47±0.26 (58.84) | 11.37±0.30 (37.80) | 12.70±0.04 (29.01) | **17.27±0.11** (28.55) |
| Full / RandLM | 0.00±0.00 (66.37) | 0.31±0.46 (8.13) | 2.24±1.28 (25.06) | **4.54±2.83** (14.57) |
| Full / FreqLM | 0.00±0.00 (68.90) | 3.36±2.79 (11.45) | 7.79±0.96 (31.50) | **8.77±0.64** (23.66) |
| Full / IterLM | 0.00±0.00 (70.37) | 2.22±1.63 (21.65) | 7.90±1.15 (35.46) | **10.69±1.34** (27.98) |
| Full / FullyLM | 8.48±0.13 (89.31) | 37.16±0.29 (76.92) | 16.32±0.08 (78.33) | **39.49±0.28** (69.00) |

Table 8: The performance (mean (%) ± std (%)) comparison of VR with XCiT-S12 on CIFAR10, and PGD-20 is used for adversarial testing. Using TRADES ($\lambda = 6$) as adversarial training. The results are presented in the format of robustness (clean accuracy), where the std of clean accuracy is omitted. The best robustness is highlighted in bold.

| In / Out | Adversarially Pre-trained Model / VR using Adversarial Training | | | |
|---|---|---|---|---|
| | Strategy 1 (✗ / ✗) | Strategy 2 (✗ / ✔) | Strategy 3 (✔ / ✗) | Strategy 4 (✔ / ✔) |
| Pad / IterLM | 0.04±0.02 (72.98) | 15.44±0.38 (58.83) | 20.81±0.06 (40.93) | **25.67±0.44** (41.78) |
| Pad / FullyLM | 0.13±0.02 (80.72) | 19.82±0.31 (63.51) | 31.84±0.01 (67.72) | **43.04±0.18** (65.01) |
| Full / IterLM | 0.12±0.08 (87.98) | 29.51±0.79 (57.09) | 23.40±0.72 (71.65) | **30.08±1.26** (52.04) |
| Full / FullyLM | 12.09±0.74 (93.84) | 48.02±0.12 (73.84) | 38.52±0.01 (91.96) | **56.31±0.07** (82.66) |

Table 9: The performance (mean (%) ± std (%)) comparison of VR with XCiT-S12 on CIFAR100, and PGD-20 is used for adversarial testing. Using TRADES ($\lambda = 6$) as adversarial training. The results are presented in the format of robustness (clean accuracy), where the std of clean accuracy is omitted. The best robustness is highlighted in bold.

| In / Out | Adversarially Pre-trained Model / VR using Adversarial Training | | | |
|---|---|---|---|---|
| | Strategy 1 (✗ / ✗) | Strategy 2 (✗ / ✔) | Strategy 3 (✔ / ✗) | Strategy 4 (✔ / ✔) |
| Pad / IterLM | 0.14±0.01 (38.54) | 3.13±0.22 (24.71) | 4.61±0.44 (11.02) | **5.74±0.14** (10.99) |
| Pad / FullyLM | 0.46±0.02 (58.32) | 11.35±0.17 (46.62) | 18.40±0.12 (44.29) | **25.77±0.03** (44.24) |
| Full / IterLM | 0.18±0.03 (57.63) | 8.42±3.58 (26.05) | **11.29±0.17** (38.07) | 6.37±0.21 (16.47) |
| Full / FullyLM | 0.78±0.13 (77.28) | 27.24±0.16 (52.63) | 22.02±0.11 (74.06) | **38.47±0.18** (64.45) |

PyTorch model repository, while the weights of the adversarially pre-trained models (ResNet-18 (A.T.), ResNet-50 (A.T.) (Salman et al., 2020) and XCiT-S12 (Debenedetti et al., 2023)) are obtained from Robustbench (Croce et al., 2021). The performance of these pre-trained models on the source domain can be found in Table 4.

## B.4 CIFAR100

We conduct experiments using the well-known image classification dataset CIFAR-100 (Krizhevsky et al., 2009) as target domain task, which consists 100 categories. The dataset consists of 60,000

Table 10: The performance (mean (%) $\pm$ std (%)) comparison of VR with XCiT-S12 on GTSRB, and PGD-20 is used for adversarial testing. Using TRADES ($\lambda = 6$) as adversarial training. The results are presented in the format of robustness (clean accuracy), where the std of clean accuracy is omitted. The best robustness is highlighted in bold.

| In / Out | Adversarially Pre-trained Model / VR using Adversarial Training | | | |
|---|---|---|---|---|
| | Strategy 1 (✘ / ✘) | Strategy 2 (✘ / ✔) | Strategy 3 (✔ / ✘) | Strategy 4 (✔ / ✔) |
| Pad / IterLM | 0.16±0.17 (47.54) | **11.36±1.05** (36.75) | 8.68±0.27 (15.83) | 8.84±0.78 (14.18) |
| Pad / FullyLM | 0.63±0.37 (65.30) | 15.74±0.05 (49.85) | 15.12±0.02 (43.36) | **25.14±0.06** (43.18) |
| Full / IterLM | 0.00±0.00 (76.03) | **41.14±0.82** (68.44) | 5.15±0.45 (41.80) | 28.70±0.02 (49.65) |
| Full / FullyLM | 1.74±0.21 (88.70) | 53.66±0.05 (84.35) | 20.85±0.25 (80.62) | **59.22±0.37** (76.97) |

Table 11: The performance (mean (%) $\pm$ std (%)) comparison of VR with ResNet-18 on CIFAR10, and PGD-20 is used for adversarial testing. Using TRADES ($\lambda = 6$) as adversarial training. The results are presented in the format of robustness (clean accuracy), where the std of clean accuracy is omitted. The best robustness is highlighted in bold.

| In / Out | Adversarially Pre-trained Model / VR using Adversarial Training | | | |
|---|---|---|---|---|
| | Strategy 1 (✘ / ✘) | Strategy 2 (✘ / ✔) | Strategy 3 (✔ / ✘) | Strategy 4 (✔ / ✔) |
| Pad / IterLM | 0.14±0.05 (58.69) | 2.93±0.23 (39.56) | 20.01±0.88 (32.49) | **20.60±0.95** (32.22) |
| Pad / FullyLM | 0.11±0.01 (71.62) | 9.50±0.20 (54.76) | 32.15±0.16 (56.70) | **37.18±0.14** (54.01) |
| Full / IterLM | 0.00±0.00 (69.22) | 0.09±0.04 (25.40) | **17.77±4.22** (58.01) | 10.41±2.22 (34.61) |
| Full / FullyLM | 0.40±0.03 (86.18) | 11.54±0.98 (55.93) | 35.20±0.21 (87.15) | **53.68±0.29** (77.94) |

Table 12: The performance (mean (%) $\pm$ std (%)) comparison of VR with ResNet-18 on CIFAR100, and PGD-20 is used for adversarial testing. Using TRADES ($\lambda = 6$) as adversarial training. The results are presented in the format of robustness (clean accuracy), where the std of clean accuracy is omitted. The best robustness is highlighted in bold.

| In / Out | Adversarially Pre-trained Model / VR using Adversarial Training | | | |
|---|---|---|---|---|
| | Strategy 1 (✘ / ✘) | Strategy 2 (✘ / ✔) | Strategy 3 (✔ / ✘) | Strategy 4 (✔ / ✔) |
| Pad / IterLM | 0.09±0.02 (20.41) | 0.44±0.26 (9.18) | **5.42±0.28** (8.96) | 5.26±0.34 (8.15) |
| Pad / FullyLM | 0.29±0.04 (46.81) | 3.49±0.15 (36.76) | 18.37±0.19 (34.24) | **21.02±0.12** (33.15) |
| Full / IterLM | 0.00±0.00 (36.87) | 0.01±0.01 (3.64) | **10.25±0.18** (31.77) | 8.20±0.14 (22.83) |
| Full / FullyLM | 0.07±0.02 (68.33) | 7.00±0.16 (39.36) | 21.91±0.12 (67.75) | **36.88±0.04** (61.32) |

$32 \times 32$ pixel color images (50,000 images for training and 10000 images for testing). We followed the same setup, taking the original training data as the training data and using the method in Appendix B.1 to generate the adversarial training data. The optimizer is AdamW. The learning rate of the optimizer is searched from set $\{1 \times 10^{-3}, 5 \times 10^{-4}\}$, while the weight decay of the optimizer is searched from set $\{10^{-2}, 10^{-3}\}$. Moreover, we use this optimizer to train pre-trained model for 60 epochs, with the learning rate decreased at 30 epochs and 50 epochs, respectively. Furthermore, in Table 6, we report the results of reprogramming a pre-trianed ResNet-18, where PGD-10 is used for adversarial training, and PGD-20 is used for adversarial testing. In Table 12, we report the results of reprogramming a pre-trianed ResNet-18, where TRADES ($\lambda = 6$) is used for adversarial training, and PGD-20 is used for adversarial testing. In Table 15, we report the results of reprogramming a pre-trianed ResNet-18, where PGD-10 under $\epsilon = 8/255$ is used for adversarial training, and PGD-20 under $\epsilon = 8/255$ is used for adversarial testing. These results also demonstrate the effectiveness of our strategy for enhancing the robustness of the reprogrammed model.

Table 13: The performance (mean (%) $\pm$ std (%)) comparison of VR with ResNet-18 on GTSRB, and PGD-20 is used for adversarial testing. Using TRADES ($\lambda = 6$) as adversarial training. The results are presented in the format of robustness (clean accuracy), where the std of clean accuracy is omitted. The best robustness is highlighted in bold.

| In / Out | Adversarially Pre-trained Model / VR using Adversarial Training | | | |
|---|---|---|---|---|
| | Strategy 1 (✗ / ✗) | Strategy 2 (✗ / ✔) | Strategy 3 (✔ / ✗) | Strategy 4 (✔ / ✔) |
| Pad / IterLM | 0.31±0.21 (37.70) | 3.17±0.85 (13.50) | 7.17±0.40 (10.30) | **7.18±0.87** (9.94) |
| Pad / FullyLM | 1.47±0.26 (58.84) | 6.99±0.27 (38.42) | 12.70±0.04 (29.01) | **16.87±0.09** (27.76) |
| Full / IterLM | 0.00±0.00 (70.37) | 0.00±0.00 (11.98) | **7.90±1.15** (35.46) | 7.67±1.15 (26.89) |
| Full / FullyLM | 8.48±0.13 (89.31) | 25.36±1.41 (64.62) | 16.32±0.08 (78.33) | **41.45±0.01** (64.62) |

Table 14: The performance (mean (%) $\pm$ std (%)) comparison of VR with ResNet-18 on CIFAR10, where PGD-10 ($\epsilon = 8/255$) is used for adversarial training, and PGD-20 ($\epsilon = 8/255$) is used for adversarial testing. The results are presented in the format of robustness (clean accuracy), where the std of clean accuracy is omitted. The best robustness is highlighted in bold.

| In / Out | Adversarially Pre-trained Model / VR using Adversarial Training | | | |
|---|---|---|---|---|
| | Strategy 1 (✗ / ✗) | Strategy 2 (✗ / ✔) | Strategy 3 (✔ / ✗) | Strategy 4 (✔ / ✔) |
| Pad / RandLM | 0.00±0.00 (56.72) | 3.96±1.31 (22.05) | 10.51±1.64 (26.59) | **13.61±0.86** (24.96) |
| Pad / FreqLM | 0.00±0.01 (58.41) | 2.33±0.87 (22.52) | 10.68±0.69 (29.69) | **14.97±1.49** (27.20) |
| Pad / IterLM | 0.00±0.00 (58.69) | 0.50±0.76 (22.30) | 10.73±0.69 (32.49) | **16.53±0.54** (30.69) |
| Pad / FullyLM | 0.00±0.00 (71.62) | 15.86±0.32 (20.49) | 13.62±0.11 (56.70) | **28.39±0.06** (49.52) |
| Full / RandLM | 0.00±0.00 (61.78) | 1.36±0.22 (25.23) | 0.22±0.14 (41.13) | **8.02±3.35** (21.44) |
| Full / FreqLM | 0.00±0.00 (70.15) | 1.57±1.35 (23.44) | 1.39±0.11 (56.05) | **6.44±0.15** (29.92) |
| Full / IterLM | 0.00±0.00 (69.22) | 2.56±1.91 (24.23) | 2.42±0.86 (58.01) | **7.62±1.11** (30.64) |
| Full / FullyLM | 0.25±0.02 (86.18) | 18.35±0.23 (32.54) | 5.37±0.13 (87.15) | **31.34±0.18** (73.43) |

Table 15: The performance (mean (%) + std (%)) comparison of VR with ResNet-18 on CIFAR100, where PGD-10 ($\epsilon = 8/255$) is used for adversarial training, and PGD-20 ($\epsilon = 8/255$) is used for adversarial testing. The results are presented in the format of robustness accuracy (clean accuracy), where the std of clean accuracy is omitted. The best robustness is highlighted in bold.

| In / Out | Adversarially Pre-trained Model / VR using Adversarial Training | | | |
|---|---|---|---|---|
| | Strategy 1 (✗ / ✗) | Strategy 2 (✗ / ✔) | Strategy 3 (✔ / ✗) | Strategy 4 (✔ / ✔) |
| Pad / RandLM | 0.00±0.00 (8.42) | 0.01±0.02 (1.20) | 0.41±0.11 (1.42) | **0.77±0.09** (1.29) |
| Pad / FreqLM | 0.00±0.00 (13.46) | 0.37±0.54 (1.02) | **1.21±0.19** (3.69) | 1.07±0.31 (2.08) |
| Pad / IterLM | 0.01±0.01 (20.41) | 0.51±0.38 (1.33) | 3.28±0.25 (8.96) | **3.74±0.28** (5.79) |
| Pad / FullyLM | 0.00±0.00 (46.81) | 2.98±0.07 (8.40) | 8.75±0.03 (34.24) | **14.77±0.13** (30.58) |
| Full / RandLM | 0.00±0.00 (12.19) | 0.27±0.41 (1.02) | 0.09±0.02 (4.40) | **0.51±0.36** (1.26) |
| Full / FreqLM | 0.00±0.00 (29.79) | 0.00±0.01 (1.37) | 1.76±0.06 (25.50) | **1.83±0.13** (18.36) |
| Full / IterLM | 0.00±0.00 (36.87) | 0.15±0.14 (2.78) | **2.48±0.11** (31.77) | 1.89±0.21 (19.85) |
| Full / FullyLM | 0.06±0.01 (68.33) | 6.62±0.09 (15.51) | 4.34±0.05 (67.75) | **20.30±0.12** (57.80) |

## B.5 CIFAR10

CIFAR-10 (Krizhevsky et al., 2009) is also a renowned image classification dataset, similar to CIFAR-100. Its data configuration is nearly the same as that of CIFAR-100 (consisting of 60,000 32

Table 16: The performance (mean (%) + std (%)) comparison of reprogrammed ResNet-18 on GT-SRB, where PGD-10 ($\epsilon = 8/255$) is used for adversarial training, and PGD-20 ($\epsilon = 8/255$) is used for adversarial testing. The results are presented in the format of robustness accuracy (clean accuracy), where the std of clean accuracy is omitted. The best robustness is highlighted in bold.

| In / Out | Adversarially Pre-trained Model / VR using Adversarial Training | | | |
|---|---|---|---|---|
| | Strategy 1 (✗ / ✗) | Strategy 2 (✗ / ✔) | Strategy 3 (✔ / ✗) | Strategy 4 (✔ / ✔) |
| Pad / RandLM | 0.03±0.04 (30.49) | 0.04±0.07 (9.20) | 3.76±0.53 (7.01) | **4.01**±**1.07** (7.21) |
| Pad / FreqLM | 0.00±0.00 (35.02) | 0.20±0.12 (10.35) | 3.68±0.67 (10.48) | **4.29**±**0.95** (9.13) |
| Pad / IterLM | 0.00±0.00 (37.70) | 3.11±2.75 (6.66) | 5.02±0.36 (10.30) | **6.74**±**0.51** (9.74) |
| Pad / FullyLM | 1.05±0.18 (58.84) | 6.05±0.28 (14.84) | 5.40±0.06 (29.01) | **12.04**±**0.09** (25.58) |
| Full / RandLM | 0.00±0.00 (66.37) | 0.01±0.01 (6.38) | 0.13±0.12 (25.06) | **1.70**±**0.98** (10.06) |
| Full / FreqLM | 0.00±0.00 (68.90) | 0.68±0.30 (6.10) | 1.26±0.23 (31.50) | **3.24**±**0.38** (16.18) |
| Full / IterLM | 0.00±0.00 (70.37) | 1.15±0.67 (8.99) | 1.77±0.32 (35.46) | **3.99**±**0.44** (20.54) |
| Full / FullyLM | 6.66±0.44 (89.31) | 24.81±0.05 (60.31) | 6.26±0.17 (78.33) | **26.90**±**0.02** (59.07) |

Table 17: The performance (mean (%) ± std (%)) comparison of VR with ResNet-18 on CIFAR10, and PGD-20 ($\epsilon = 8/255$) is used for adversarial testing. Using TRADES ($\lambda = 6$ and $\epsilon = 8/255$) as adversarial training. The results are presented in the format of robustness (clean accuracy), where the std of clean accuracy is omitted. The best robustness is highlighted in bold.

| In / Out | Adversarially Pre-trained Model / VR using Adversarial Training | | | |
|---|---|---|---|---|
| | Strategy 1 (✗ / ✗) | Strategy 2 (✗ / ✔) | Strategy 3 (✔ / ✗) | Strategy 4 (✔ / ✔) |
| Pad / IterLM | 0.00±0.00 (58.69) | 0.61±0.03 (36.79) | 10.73±0.69 (32.49) | **14.42**±**0.96** (33.30) |
| Pad / FullyLM | 0.00±0.00 (71.62) | 2.28±0.20 (50.26) | 13.62±0.11 (56.70) | **25.44**±**0.06** (50.62) |
| Full / IterLM | 0.00±0.00 (69.22) | 0.02±0.04 (24.69) | 2.42±0.86 (58.01) | **4.33**±**0.67** (30.02) |
| Full / FullyLM | 0.25±0.02 (86.18) | 5.37±0.80 (46.68) | 5.37±0.13 (87.15) | **30.43**±**0.04** (71.11) |

× 32 pixel color images, with 50,000 images for training and 10,000 images for testing), except that CIFAR-10 has only 10 classification categories. We use the same optimizer as for CIFAR-100 for reprogramming. In Table 5, we report the results of reprogramming a pre-trianed ResNet-18, where PGD-10 is used for adversarial training, and PGD-20 is used for adversarial testing. In Table 11, we report the results of reprogramming a pre-trianed ResNet-18, where TRADES ($\lambda = 6$) is used for adversarial training, and PGD-20 is used for adversarial testing. In Table 14, we report the results of reprogramming a pre-trianed ResNet-18, where PGD-10 under $\epsilon = 8/255$ is used for adversarial training, and PGD-20 under $\epsilon = 8/255$ is used for adversarial testing. These results also indicate that incorporating adversarial samples from the target domain and using adversarially pre-trained models can both enhance the robustness of the reprogrammed model. In addition, we present some visual results of the reprogramming in Figures 4 and 5.

## B.6 GTSRB

The German Traffic Sign Recognition Benchmark (GTSRB) (Stallkamp et al., 2012) contains 43 classes of traffic signs, split into 39,209 training images and 12,630 test images. Due to the smaller image size and fewer classification categories compared with IMAGENET-1K, we use this dataset as our target domain task for reprogramming. The images have varying light conditions and rich backgrounds. We continue to use their data partitioning for training and testing. We use the same optimizer as for CIFAR-100 for reprogramming. In Table 7, we report the results of reprogramming a pre-trianed ResNet-18, where PGD-10 is used for adversarial training, and PGD-20 is used for adversarial testing. In Table 13, we report the results of reprogramming a pre-trianed ResNet-18, where TRADES ($\lambda = 6$) is used for adversarial training, and PGD-20 is used for adversarial testing. In Table 14, we report the results of reprogramming a pre-trianed ResNet-18, where PGD-10 under

Table 18: The performance (mean (%) $\pm$ std (%)) comparison of VR with ResNet-18 on CIFAR100, and PGD-20 ($\epsilon = 8/255$) is used for adversarial testing. Using TRADES ($\lambda = 6$ and $\epsilon = 8/255$) as adversarial training. The results are presented in the format of robustness (clean accuracy), where the std of clean accuracy is omitted. The best robustness is highlighted in bold.

| In / Out | Adversarially Pre-trained Model / VR using Adversarial Training | | | |
| --- | --- | --- | --- | --- |
| | Strategy 1 (✘ / ✘) | Strategy 2 (✘ / ✔) | Strategy 3 (✔ / ✘) | Strategy 4 (✔ / ✔) |
| Pad / IterLM | 0.01±0.01 (20.41) | 0.05±0.03 (7.95) | 3.28±0.25 (8.96) | **3.66±0.09** (8.09) |
| Pad / FullyLM | 0.00±0.00 (46.81) | 0.49±0.09 (34.64) | 8.75±0.03 (34.24) | **13.36±0.04** (31.97) |
| Full / IterLM | 0.00±0.00 (36.87) | 0.00±0.00 (3.38) | **2.48±0.11** (31.77) | 0.55±0.07 (4.57) |
| Full / FullyLM | 0.06±0.01 (68.33) | 2.17±0.03 (31.97) | 4.34±0.05 (67.75) | **19.02±0.07** (56.02) |

Table 19: The performance (mean (%) $\pm$ std (%)) comparison of VR with ResNet-18 on GTSRB, and PGD-20 ($\epsilon = 8/255$) is used for adversarial testing. Using TRADES ($\lambda = 6$ and $\epsilon = 8/255$) as adversarial training. The results are presented in the format of robustness (clean accuracy), where the std of clean accuracy is omitted. The best robustness is highlighted in bold.

| In / Out | Adversarially Pre-trained Model / VR using Adversarial Training | | | |
| --- | --- | --- | --- | --- |
| | Strategy 1 (✘ / ✘) | Strategy 2 (✘ / ✔) | Strategy 3 (✔ / ✘) | Strategy 4 (✔ / ✔) |
| Pad / IterLM | 0.00±0.00 (37.70) | 0.47±0.06 (11.64) | 5.02±0.36 (10.30) | **6.02±0.91** (10.52) |
| Pad / FullyLM | 1.05±0.18 (58.84) | 0.93±0.07 (34.88) | 5.40±0.06 (29.01) | **10.46±0.04** (25.70) |
| Full / IterLM | 0.00±0.00 (70.37) | **3.80±3.29** (6.33) | 1.77±0.32 (35.46) | 2.03±1.12 (17.37) |
| Full / FullyLM | 6.66±0.44 (89.31) | 12.85±0.64 (57.77) | 6.26±0.17 (78.33) | **28.74±0.16** (55.32) |

$\epsilon = 8/255$ is used for adversarial training, and PGD-20 under $\epsilon = 8/255$ is used for adversarial testing. These results also validate our strategy.

### B.7 ADVERSARIAL TRAINING WITH LARGER PERTURBATION RADIUS

We keep other settings unchanged and set the perturbation radius for adversarial training and testing to $\epsilon = 8/255$ to further validate our analytical results. Tables 14 to 19 present the results of ResNet-18 on different datasets. We observe that these results exhibit slightly lower robustness, which may be attributed to the relatively high attack intensity at $\epsilon = 8/255$. Furthermore, as shown in Table 4, although the adversarially pre-trained model demonstrates higher robustness under $\epsilon = 4/255$, its robustness significantly degrades when facing attacks at $\epsilon = 8/255$. This leads to excessively high adversarial risk for the pre-trained model in the source domain. According to our theoretical analysis, this elevated risk may result in increased adversarial risk in the target domain after reprogramming. Nevertheless, these results align well with our analysis.

## C EXPERIMENTS ON COMPLEX MODELS AND LARGE DATASETS

To further validate the effectiveness of utilizing adversarially pre-trained models and employing adversarial training during the reprogramming process, we conducte experiments using adversarially pre-trained models on ImageNet, including XCiT-S12 and ViT-S+ConvStem (Singh et al., 2023), which feature more complex architectures compared with ResNet-18 and ResNet-50. Additionally, we perform experiments on larger datasets, including Flowers102 (Nilsback & Zisserman, 2008), OxfordPets (Parkhi et al., 2012), SUN397 (Xiao et al., 2010), and Food101 (Bossard et al., 2014), which offer higher image resolutions compared with CIFAR-10, CIFAR-100, and GTSRB. Moreover, SUN397 and Food101 provide more extensive training and validation data. Following the setup in (Chen et al., 2023b), we resize the images to a resolution of $128 \times 128$, while keeping all other experimental settings consistent with those described above.

Table 20: The performance (mean (%) + std (%)) comparison of reprogrammed XCiT-S12 on Flowers102, where PGD-10 is used for adversarial training, and PGD-20 is used for adversarial testing. The results are presented in the format of robustness accuracy (clean accuracy), where the std of clean accuracy is omitted. The best robustness is highlighted in bold.

| In / Out | Adversarially Pre-trained Model / VR using Adversarial Training | | | |
| --- | --- | --- | --- | --- |
| | Strategy 1 (✘ / ✘) | Strategy 2 (✘ / ✔) | Strategy 3 (✔ / ✘) | Strategy 4 (✔ / ✔) |
| Pad / IterLM | 0.00±0.00 (6.64) | 0.00±0.00 (6.40) | 1.83±0.24 (6.53) | **1.86±0.22** (6.57) |
| Pad / FullyLM | 0.05±0.07 (66.60) | 0.00±0.00 (21.26) | 29.06±0.08 (64.47) | **37.16±0.48** (66.25) |
| Full / IterLM | 0.00±0.00 (6.90) | 0.00±0.00 (5.85) | 0.86±0.07 (6.20) | **1.53±0.30** (5.83) |
| Full / FullyLM | 0.00±0.00 (65.58) | 0.04±0.06 (26.85) | 26.46±0.94 (70.17) | **39.93±0.30** (72.35) |

Table 21: The performance (mean (%) + std (%)) comparison of reprogrammed XCiT-S12 on OxfordPets, where PGD-10 is used for adversarial training, and PGD-20 is used for adversarial testing. The results are presented in the format of robustness accuracy (clean accuracy), where the std of clean accuracy is omitted. The best robustness is highlighted in bold.

| In / Out | Adversarially Pre-trained Model / VR using Adversarial Training | | | |
| --- | --- | --- | --- | --- |
| | Strategy 1 (✘ / ✘) | Strategy 2 (✘ / ✔) | Strategy 3 (✔ / ✘) | Strategy 4 (✔ / ✔) |
| Pad / IterLM | 0.00±0.00 (70.94) | 0.04±0.02 (38.90) | 30.82±0.61 (61.23) | **30.85±0.57** (59.74) |
| Pad / FullyLM | 0.01±0.02 (86.37) | 0.36±0.51 (5.19) | 28.32±0.08 (77.37) | **37.23±0.02** (78.00) |
| Full / IterLM | 0.00±0.00 (72.01) | 0.17±0.04 (37.33) | 28.61±0.02 (66.87) | **41.02±0.36** (67.37) |
| Full / FullyLM | 0.00±0.00 (87.63) | 4.00±0.06 (35.97) | 38.07±0.14 (85.90) | **47.14±0.18** (86.55) |

Table 22: The performance (mean (%) + std (%)) comparison of reprogrammed XCiT-S12 on SUN397, where PGD-10 is used for adversarial training, and PGD-20 is used for adversarial testing. The results are presented in the format of robustness accuracy (clean accuracy), where the std of clean accuracy is omitted. The best robustness is highlighted in bold.

| In / Out | Adversarially Pre-trained Model / VR using Adversarial Training | | | |
| --- | --- | --- | --- | --- |
| | Strategy 1 (✘ / ✘) | Strategy 2 (✘ / ✔) | Strategy 3 (✔ / ✘) | Strategy 4 (✔ / ✔) |
| Pad / IterLM | 0.00±0.01 (30.49) | 2.19±0.03 (2.19) | **5.63±0.05** (17.27) | 5.42±0.14 (10.85) |
| Pad / FullyLM | 0.18±0.13 (59.19) | 2.38±0.30 (2.80) | 8.68±0.37 (49.90) | **22.32±0.08** (51.07) |
| Full / IterLM | 0.00±0.00 (25.97) | 2.19±0.03 (2.19) | 4.88±0.10 (20.99) | **5.72±0.31** (16.27) |
| Full / FullyLM | 0.23±0.11 (59.31) | 6.71±0.07 (20.02) | 7.97±0.64 (56.30) | **24.31±0.12** (58.49) |

## C.1 Description of Datasets

**Flower102** comprises 102 flower species commonly found in the United Kingdom, with each category containing between 40 and 258 images. The dataset exhibits significant variations in scale, pose, and lighting conditions, as well as intra-class diversity and inter-class similarity. It is divided into training, validation, and test sets, with the training and validation sets each containing 10 images per class (1,020 images in total), while the test set includes the remaining 6,149 images (at least 20 images per class). **OxfordPets** consists of 37 pet categories, each represented by approximately 200 images, and is split into 3,680 training and 3,669 test images. **SUN397**, designed for scene understanding research, contains 108,753 images spanning 397 categories, with each category including a minimum of 100 images. A random 75% of the data is allocated for training, with the remaining 25% reserved for testing. Finally, **Food101** includes 101 food categories with a total of 101,000 images. Each category comprises 750 uncleaned training images and 250 manually curated

Table 23: The performance (mean (%) + std (%)) comparison of reprogrammed XCiT-S12 on Food101, where PGD-10 is used for adversarial training, and PGD-20 is used for adversarial testing. The results are presented in the format of robustness accuracy (clean accuracy), where the std of clean accuracy is omitted. The best robustness is highlighted in bold.

| In / Out | Adversarially Pre-trained Model / VR using Adversarial Training | | | |
| --- | --- | --- | --- | --- |
| | Strategy 1 (✘ / ✘) | Strategy 2 (✘ / ✔) | Strategy 3 (✔ / ✘) | Strategy 4 (✔ / ✔) |
| Pad / IterLM | 0.00±0.00 (24.56) | 0.24±0.31 (1.00) | **2.61±0.11** (11.45) | 1.56±0.36 (4.96) |
| Pad / FullyLM | 0.03±0.02 (67.00) | 0.83±0.07 (1.00) | 9.79±0.03 (50.35) | **21.58±0.03** (45.72) |
| Full / IterLM | 0.00±0.00 (19.32) | **0.93±0.05** (1.02) | 0.74±0.06 (10.26) | 0.61±0.51 (1.07) |
| Full / FullyLM | 0.05±0.01 (64.69) | 5.82±0.33 (12.30) | 9.27±0.03 (57.77) | **23.26±0.04** (52.80) |

Table 24: The performance (mean (%) + std (%)) comparison of reprogrammed ViT-S+ConvStem on Flowers102, where PGD-10 is used for adversarial training, and PGD-20 is used for adversarial testing. The results are presented in the format of robustness accuracy (clean accuracy), where the std of clean accuracy is omitted. The best robustness is highlighted in bold.

| In / Out | Adversarially Pre-trained Model / VR using Adversarial Training | | | |
| --- | --- | --- | --- | --- |
| | Strategy 1 (✘ / ✘) | Strategy 2 (✘ / ✔) | Strategy 3 (✔ / ✘) | Strategy 4 (✔ / ✔) |
| Pad / IterLM | 0.00±0.00 (8.72) | 0.00±0.00 (3.12) | 2.31±0.05 (7.01) | **2.34±0.05** (7.06) |
| Pad / FullyLM | 0.02±0.02 (55.20) | 0.76±0.02 (33.06) | 33.72±0.18 (63.66) | **39.67±0.31** (64.44) |
| Full / IterLM | 0.00±0.00 (7.12) | 0.00±0.00 (4.36) | **2.04±0.15** (8.81) | 1.77±0.07 (7.15) |
| Full / FullyLM | 0.00±0.00 (68.47) | 3.11±1.27 (50.68) | 34.91±0.25 (68.42) | **44.08±0.09** (71.69) |

Table 25: The performance (mean (%) + std (%)) comparison of reprogrammed ViT-S+ConvStem on OxfordPets, where PGD-10 is used for adversarial training, and PGD-20 is used for adversarial testing. The results are presented in the format of robustness accuracy (clean accuracy), where the std of clean accuracy is omitted. The best robustness is highlighted in bold.

| In / Out | Adversarially Pre-trained Model / VR using Adversarial Training | | | |
| --- | --- | --- | --- | --- |
| | Strategy 1 (✘ / ✘) | Strategy 2 (✘ / ✔) | Strategy 3 (✔ / ✘) | Strategy 4 (✔ / ✔) |
| Pad / IterLM | 0.00±0.00 (60.73) | 0.00±0.00 (29.17) | 34.99±0.16 (59.24) | **35.59±0.65** (59.58) |
| Pad / FullyLM | 0.00±0.00 (75.13) | 0.21±0.02 (39.65) | 35.82±0.31 (77.10) | **39.88±0.24** (76.47) |
| Full / IterLM | 0.00±0.00 (70.33) | 0.15±0.02 (3.66) | 34.84±0.45 (64.69) | **44.00±0.12** (65.56) |
| Full / FullyLM | 0.00±0.00 (85.48) | 11.29±0.10 (53.93) | 43.23±0.19 (82.82) | **49.33±0.12** (83.04) |

test images. The training images contain a degree of noise, including intense colors and occasional mislabeling, while all images are rescaled to a maximum side length of 512 pixels.

## C.2 EXPERIMENTAL RESULTS

We present these experimental results in Tables 20 to 27, which demonstrate the effectiveness of our strategy on more complex models as well as on larger datasets.

## C.3 EXPERIMENTS ON MEDICAL DATASETS

We also conduct experimental on the ChestX-Ray (Govi, 2020) benchmark dataset (5,933 2D radiographs: 5,309 training/624 testing) containing healthy subjects and six etiological pulmonary infection subtypes, demonstrating our strategy's effectiveness in safety-critical medical imaging diagnostics. The results in Table 28 remain consistent with our main conclusions.

Table 26: The performance (mean (%) + std (%)) comparison of reprogrammed ViT-S+ConvStem on SUN397, where PGD-10 is used for adversarial training, and PGD-20 is used for adversarial testing. The results are presented in the format of robustness accuracy (clean accuracy), where the std of clean accuracy is omitted. The best robustness is highlighted in bold.

| In / Out | Adversarially Pre-trained Model / VR using Adversarial Training | | | |
| --- | --- | --- | --- | --- |
| | Strategy 1 (✘ / ✘) | Strategy 2 (✘ / ✔) | Strategy 3 (✔ / ✘) | Strategy 4 (✔ / ✔) |
| Pad / IterLM | 0.02±0.00 (23.41) | 1.22±1.30 (2.06) | 8.56±0.16 (20.59) | **9.91±0.29** (20.20) |
| Pad / FullyLM | 0.00±0.01 (52.16) | 4.04±0.13 (15.42) | 11.92±0.13 (51.51) | **24.17±0.12** (53.50) |
| Full / IterLM | 0.00±0.00 (27.29) | 2.19±0.03 (2.19) | 6.71±0.33 (21.82) | **8.63±0.01** (19.06) |
| Full / FullyLM | 0.01±0.00 (61.01) | 12.56±0.15 (41.11) | 11.33±0.05 (54.70) | **26.03±0.04** (67.71) |

Table 27: The performance (mean (%) + std (%)) comparison of reprogrammed ViT-S+ConvStem on Food101, where PGD-10 is used for adversarial training, and PGD-20 is used for adversarial testing. The results are presented in the format of robustness accuracy (clean accuracy), where the std of clean accuracy is omitted. The best robustness is highlighted in bold.

| In / Out | Adversarially Pre-trained Model / VR using Adversarial Training | | | |
| --- | --- | --- | --- | --- |
| | Strategy 1 (✘ / ✘) | Strategy 2 (✘ / ✔) | Strategy 3 (✔ / ✘) | Strategy 4 (✔ / ✔) |
| Pad / IterLM | 0.00±0.00 (16.23) | 0.51±0.26 (0.99) | 4.04±0.10 (12.27) | **4.30±0.05** (11.24) |
| Pad / FullyLM | 0.00±0.00 (56.99) | 4.36±0.05 (13.35) | 12.29±0.17 (48.74) | **21.23±0.02** (45.54) |
| Full / IterLM | 0.00±0.00 (15.62) | 0.25±0.23 (1.71) | 0.84±0.04 (12.50) | **0.98±0.00** (1.00) |
| Full / FullyLM | 0.00±0.00 (67.94) | 12.41±0.05 (31.70) | 10.91±0.02 (53.47) | **23.15±0.07** (50.34) |

Table 28: The performance (mean (%) + std (%)) comparison of reprogrammed ResNet-18 on ChestX-Ray, where PGD-10 is used for adversarial training, and PGD-20 is used for adversarial testing. The results are presented in the format of robustness accuracy (clean accuracy), where the std of clean accuracy is omitted. The best robustness is highlighted in bold.

| In / Out | Adversarially Pre-trained Model / VR using Adversarial Training | | | |
| --- | --- | --- | --- | --- |
| | Strategy 1 (✘ / ✘) | Strategy 2 (✘ / ✔) | Strategy 3 (✔ / ✘) | Strategy 4 (✔ / ✔) |
| Pad / IterLM | 0.00±0.00 (70.31) | 32.23±1.69 (56.30) | 43.95±0.23 (66.21) | **49.41±0.79** (64.26) |
| Pad / FullyLM | 21.68±0.87 (72.07) | 52.61±0.37 (64.23) | 46.29±0.10 (71.68) | **55.86±0.04** (70.51) |
| Full / IterLM | 10.74±0.22 (74.80) | 50.30±1.21 (54.30) | 33.59±0.45 (67.77) | **52.34±0.15** (65.04) |
| Full / FullyLM | 40.23±0.16 (74.61) | 54.30±0.31 (66.79) | 9.57±0.05 (71.48) | **55.86±0.11** (71.88) |

## C.4 EXPERIMENTS UNDER DIFFERENT ADVERSARIAL ATTACKS

To further verify the effectiveness of our strategy in a more comprehensive way, we verify the robustness under more adversarial attacks, including FGSM (Goodfellow et al., 2014), Square (Andriushchenko et al., 2020) and AutoAttack (Croce & Hein, 2020). The experimental results are shown in Table 29. These results show that different adversarial attack methods will bring different adversarial robustness results, and our strategy can achieve satisfactory adversarial robustness on all these methods.

Table 29: The performance (mean (%) + std (%)) comparison of VR (Full / FullyLM) with ViT-S+ConvStem (A.T.) on different datasets under different adversarial attacks. Using PGD-10 ($\epsilon = 4/255$) as adversarial training.

| Dataset | Clean Acc. | FGSM | PGD-20 | Square | AutoAttack |
|---|---|---|---|---|---|
| Flowers102 | 71.69±0.24 | 47.18±0.12 | 44.08±0.09 | 60.42±0.03 | 40.89±0.07 |
| OxfordPets | 83.04±0.18 | 54.36±0.26 | 49.33±0.12 | 70.81±0.08 | 48.42±0.14 |
| SUN397 | 67.17±3.49 | 38.50±0.61 | 26.03±0.04 | 55.95±0.69 | 30.02±0.66 |
| Food101 | 50.34±0.13 | 24.88±0.10 | 23.15±0.07 | 34.88±0.03 | 19.26±0.15 |

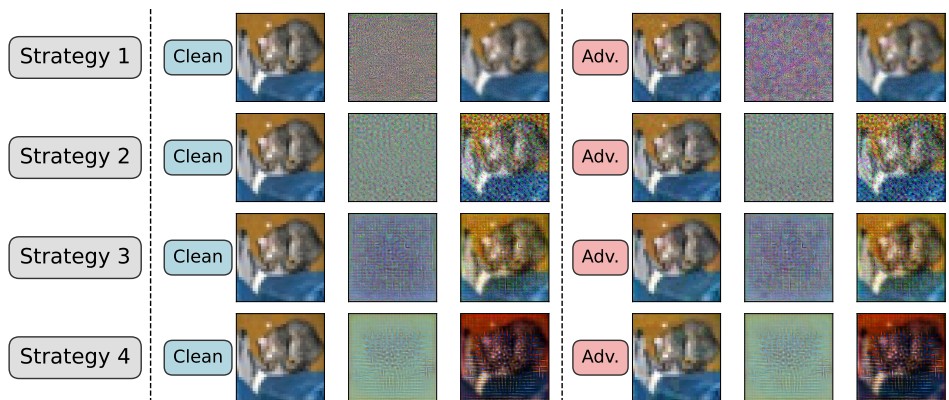

Figure 4: Visual results of reprogramming a ResNet-18 on the CIFAR-10 dataset, where the visual input transformation used for reprogramming is Full and the output label mapping is FullyLM. For each group of images, the leftmost image is the original image from the target domain, the middle image shows the noise added by the visual input transformation, and the rightmost image is the output after applying the visual input transformation.

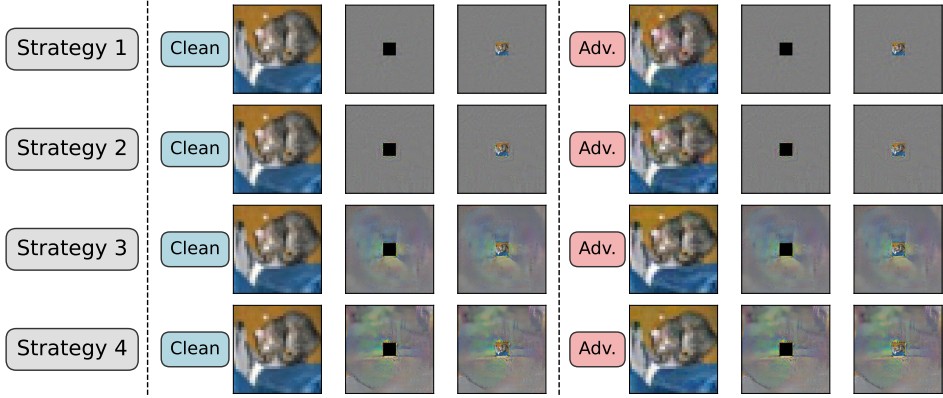

Figure 5: Visual results of reprogramming a ResNet-18 on the CIFAR-10 dataset, where the visual input transformation used for reprogramming is Pad and the output label mapping is FullyLM. For each group of images, the leftmost image is the original image from the target domain, the middle image shows the noise added by the visual input transformation, and the rightmost image is the output after applying the visual input transformation.

