# OpenReview forum: "Endowing Visual Reprogramming with Adversarial Robustness"
_ICLR.cc/2025/Conference — ICLR 2025 Poster_

### Official Review · Reviewer_R93c · 2024-11-02

**Soundness:** 3
**Presentation:** 3
**Contribution:** 2
**Rating:** 6
**Confidence:** 4

**Summary:**

This paper investigates the adversarial robustness of visual reprogramming (VR), a method that adapts pre-trained models for new tasks without fine-tuning. The authors find that existing VR methods are vulnerable to adversarial attacks, but can be significantly improved by using adversarially robust pre-trained models and incorporating adversarial examples from the target task during reprogramming. The paper presents a theoretical analysis that bounds the adversarial risk of VR and provides a foundation for future research. Experiments demonstrate the effectiveness of the proposed strategies in enhancing the robustness of reprogrammed models.

**Strengths:**

- Pioneering exploration of VR's adversarial robustness: This paper is the first to focus on the vulnerability of VR models when facing adversarial attacks and proposes corresponding solutions, filling a gap in this field.
- The article not only experimentally verifies that using adversarial pre-trained models and adversarial samples can improve the adversarial robustness of VR models but also proposes a theoretical guarantee of an adversarial robustness risk upper bound, providing a theoretical foundation for future research.

**Weaknesses:**

- I am concerned about the application scenarios of the proposed method. Visual reprogramming aims to leverage less data and computational resource overhead to stimulate the base model's capabilities on target tasks, which means that timeliness and performance are more important than adversarial robustness. The methods proposed in the article hinder these two aspects to some extent.
- The paper primarily uses the projected gradient descent (PGD) method to generate adversarial samples. However, different generation methods may have different impacts on the model's robustness, and there is a lack of discussion on more adversarial sample generation methods.
- This work focuses on classification accuracy on adversarial samples, but adversarial robustness is a multidimensional concept that requires more comprehensive evaluation metrics. Adversarial training has some limitations, such as high computational costs and the risk of overfitting, which require more discussion.

**Questions:**

The paper focuses on image classification tasks. VR can be applied to a wider range of tasks, such as object detection and semantic segmentation. Can the methods in this article be extended to other downstream tasks?

---

> ### Author Response · Authors · 2024-11-21
> **Reply to Reviewer R93c (Part 1/3)**
>
> Thank you for your expertise and attention to detail that has been instrumental in improving the quality of my work. We are delighted that you consider our research to be effective, and interesting. Our point-by-point responses to the reviewer's mentioned questions (Q) and weaknesses (W) are provided as follows.
>
> **W1: I am concerned about the application scenarios of the proposed method. Visual reprogramming aims to leverage less data and computational resource overhead to stimulate the base model's capabilities on target tasks, which means that timeliness and performance are more important than adversarial robustness. The methods proposed in the article hinder these two aspects to some extent.**
>
> We greatly appreciate your concerns regarding the application scenarios of Visual Reprogramming (VR). Here, we aim to clarify the design motivations behind our work and the importance of adversarial robustness. In certain critical application scenarios, such as medical diagnosis, autonomous driving, and financial transactions, the security of the target task is paramount. In these cases, the adversarial robustness of the model is directly tied to the reliability and safety of the overall system. Therefore, adversarial robustness is an indispensable requirement in such applications, as important as performance such as accuracy.
>
> While existing VR research has achieved promising results, it generally lacks an in-depth exploration of adversarial robustness. Our research seeks to address this gap by providing baseline results in this area, offering theoretical insights and experimental evidence to inspire future research.
>
> Currently, adversarial training is the most effective and widely adopted method to enhance the adversarial robustness of models. Therefore, to effectively improve the adversarial robustness of VR, we conduct experiments involving adversarial training. However, this does not diminish the inherent advantages of VR. For example, since VR does not require modifications to the pre-trained model and involves only a small number of trainable parameters, employing adversarial training in VR can still reduce computational resource requirements compared to training a model from scratch. This makes it possible to obtain a larger robust model within the same computational budget.
>
> Regarding the impact on timeliness and performance, these are well-recognized challenges in adversarial training research and are also present in VR. In our experiments, we adopt PGD-AT and TRADES as they are two classical and effective adversarial training methods. Our primary goal is to demonstrate that adversarial training is an effective approach to improving the robustness of VR. We believe that designing better adversarial training algorithms tailored for VR could further enhance adversarial robustness and performance while reducing computational resource consumption.
>
> Moreover, our experimental results demonstrate that even without adversarial training, utilizing adversarially pre-trained models can provide certain robustness. For example, as shown in the table below:
>
> Table 1: The performance comparison of VR with XCIT-S12 (A.T.) on different datasets, and PGD-20 is used for adversarial testing. Using PGD-10 ($\epsilon=4/255$) as adversarial training. The results are presented in the format of robustness (clean accuracy). The best robustness is highlighted in bold.
>
> |    In / Out    |    Flowers102     |    OxfordPets     |      SUN397       |      Food101      |
> | :------------: | :---------------: | :---------------: | :---------------: | :---------------: |
> | Pad / FullyLM  |   37.99 (68.21)   |   36.61 (77.85)   |   21.67 (51.03)   |   21.47 (45.79)   |
> | Full / FullyLM | **40.88 (74.77)** | **46.81 (86.44)** | **23.99 (58.09)** | **23.25 (52.77)** |
>
> Table 2: The performance of VR with XCIT-S12 (A.T.) under natural training (N.T.) on different datasets, and PGD-20 is used for adversarial testing. The results are presented in the format of robustness (clean accuracy). The best robustness is highlighted in bold.
>
> |    In / Out    |    Flowers102     |    OxfordPets     |      SUN397      |     Food101      |
> | :------------: | :---------------: | :---------------: | :--------------: | :--------------: |
> | Pad / FullyLM  | **29.47 (65.66)** |   28.35 (77.29)   | **7.80 (48.89)** | **9.81 (50.10)** |
> | Full / FullyLM |   28.89 (77.31)   | **38.03 (85.88)** |   7.23 (54.96)   |   9.32 (57.74)   |
>
> Continue by the next post.

---

> ### Author Response · Authors · 2024-11-21
> **Reply to Reviewer R93c (Part 2/3)**
>
> When applying an adversarially pre-trained ViT on ImageNet to the OxfordPets target dataset, adversarial training achieves 49.41\% robustness, while non-adversarial training only decreases by 6.33\% robustness while maintaining almost the same accuracy. These findings highlight the potential of leveraging adversarial pre-training as a complementary approach to improve robustness in visual reprogramming tasks, offering a practical solution for safety scenarios where training costs need to be reduced. We believe this provides valuable insights and a strong foundation for future research aimed at enhancing both robustness and efficiency in VR applications.
>
> **W2: The paper primarily uses the projected gradient descent (PGD) method to generate adversarial samples. However, different generation methods may have different impacts on the model's robustness, and there is a lack of discussion on more adversarial sample generation methods.**
>
> Regarding adversarial training methods, we use PGD-AT and TRADES as the primary methods for generating adversarial samples during adversarial training. This is because these two methods are well-established and effective adversarial training approaches commonly employed in the literature, and our primary goal is to demonstrate that adversarial training is an effective approach to improving the robustness of VR. We believe that employing more advanced adversarial sample generation methods could potentially lead to improved results. For example, in VR, methods such as AVmixup [1] or DAJAT [2] could be employed to generate additional adversarial examples under various augmentation strategies, potentially further enhancing the robust generalization ability of VR. Additionally, approaches like MART [3] and SCORE [4], which refine the objective of adversarial training, could be utilized to better balance the accuracy and robustness of VR.
>
> [1] Saehyung Lee, Hyungyu Lee, and Sungroh Yoon. Adversarial vertex mixup: Toward better adversarially robust generalization, 2020b.
>
> [2] Sravanti Addepalli, Samyak Jain, and R. Venkatesh Babu. Efficient and effective augmentation strategy for adversarial training, 2022.
>
> [3] Yisen Wang, Difan Zou, Jinfeng Yi, James Bailey, Xingjun Ma, and Quanquan Gu. Improving adversarial robustness requires revisiting misclassified examples. In ICLR, 2020.
>
> [4] Tianyu Pang, Min Lin, Xiao Yang, Jun Zhu, and Shuicheng Yan. Robustness and accuracy could be reconcilable by (proper) definition, 2022.
>
> **W3: This work focuses on classification accuracy on adversarial samples, but adversarial robustness is a multidimensional concept that requires more comprehensive evaluation metrics. Adversarial training has some limitations, such as high computational costs and the risk of overfitting, which require more discussion.**
>
> We agree that adversarial robustness is a multidimensional concept. According to your suggestion, we include the following additional adversarial robustness evaluation methods to further evaluate the adversarial robustness of VR: CW, Square, and AutoAttack. We conduct additional experiments using these methods to evaluate the robustness of VR, and the results are presented in the table below:
>
> Table 3: The performance comparison of VR (Full / FullyLM) with ViT-S+ConvStem (A.T.) on different datasets under different adversarial attacks. Using PGD-10 ($\epsilon=4/255$) as adversarial training. The results are presented in the format of robustness (clean accuracy).
>
> |  Dataset   | Clean Acc. | PGD-20 |  CW   | Square | AutoAttack |
> | :--------: | :--------: | :----: | :---: | :----: | :--------: |
> | Flowers102 |   71.68    | 44.28  | 43.98 | 60.45  |   40.90    |
> | OxfordPets |   82.95    | 49.41  | 52.57 | 70.98  |   48.49    |
> |   SUN397   |   57.57    | 25.84  | 29.53 | 44.69  |   23.12    |
> |  Food101   |   50.38    | 23.36  | 29.57 | 35.29  |   19.45    |
>
> From the results, it can be observed that our strategy is robust under various types of adversarial attacks.
>
> Continue by the next post.

---

> ### Author Response · Authors · 2024-11-21
> **Reply to Reviewer R93c (Part 3/3)**
>
> **Q1: The paper focuses on image classification tasks. VR can be applied to a wider range of tasks, such as object detection and semantic segmentation. Can the methods in this article be extended to other downstream tasks?**
>
> Our method can indeed be extended to other downstream tasks beyond image classification. While the primary focus of this work is on image classification, as outlined in [IBM/model-reprogramming](https://github.com/IBM/model-reprogramming), the proposed approach can be adapted to different tasks by designing appropriate input transformations and output label mappings, depending on the specific source and target domains.
>
> For example, [1] has demonstrated the vulnerability of semantic segmentation models to adversarial attacks. Building on this, one can leverage an adversarially pretrained vision model to achieve a robust semantic segmentation model by VR. Alternatively, by utilizing existing robust semantic segmentation models, as shown in [2], our method can be applied to adapt these models to new robust semantic segmentation tasks.
>
> [1] Volker Fischer, Mummadi Chaithanya Kumar, Jan Hendrik Metzen, and Thomas Brox. Adversarial examples for semantic image segmentation, 2017
>
> [2] Jindong Gu, Hengshuang Zhao, Volker Tresp, and Philip Torr. Segpgd: An effective and efficient adversarial attack for evaluating and boosting segmentation robustness, 2023.
>
> We hope that our responses can satisfactorily address your concerns. Thank you again for your time to provide us with valuable feedback.

---

> ### Author Response · Authors · 2024-11-24
> **Looking forward to your response**
>
> We genuinely appreciate your thorough review and constructive comments on our manuscript. Your feedback has provided us with valuable insights to help us further improve our work. We hope that our rebuttal has addressed your concerns. If you have further questions and suggestions, we would be happy to continue the discussion with you! Your participation in our work is greatly appreciated.

---

> > ### Comment · Reviewer_R93c · 2024-11-26
> >
> > Sorry for the late reply. I appreciate the authors’ efforts in responding to comments and revising the manuscript. My concerns have been largely addressed. Additionally, I hope that, if possible, the authors can provide more experiments in the final version that demonstrate practical application value, such as those mentioned in the response regarding medical diagnosis, finance, or autonomous driving. I am willing to raise my score to 6.

---

> > > ### Author Response · Authors · 2024-11-26
> > > **Thanks for your feedback!**
> > >
> > > Thank you so much for recognizing our contributions! We sincerely appreciate your valuable comments and constructive suggestions, which definitely have helped us improve the quality of our paper. We will provide more experiments in the final version that demonstrate practical application value. Thank you again for your willingness to raise the score and your recognition of our work.

---

### Official Review · Reviewer_krkJ · 2024-11-03

**Soundness:** 3
**Presentation:** 3
**Contribution:** 3
**Rating:** 6
**Confidence:** 3

**Summary:**

The goal of Visual reprogramming is re-purposing pre-trained models for new tasks without retraining the model itself, and only learning a lightweight visual input transformation and label remapping of the output. Prior works have used different kinds of input transforms, remapping functions, losses etc, while this work focuses on the adversarial robustness aspect. The authors find that reprogramming naturally trained models results in almost zero robustness against adversarial attacks. Empirically, the paper demonstrates that using adversarially pre-trained models as the backbone for reprogramming and using adversarial training during the reprogramming both improve robustness. Combining both strategies (adversarially pre-trained backbones + adversarial training) yields the best results. The paper also provides some theoretical lemmas and proofs to establish an upper bound for adversarial risk of visual reprogramming.

**Strengths:**

+ The paper is easy to follow

+ The work explores different combinations of input transformations and label remapping techniques, which strengthens confidence in the results.

**Weaknesses:**

- Limited novelty, the main component of the method is using adversarially pre-trained backbones, and using adversarial training during visual re-programming, which is a direct application of prior work.

- The theoretical portion of the paper tries to tackle an interesting question, however the relevance to the method is limited ? The method itself is very empirical and intuitive, and the theory seems like an afterthought.

- Though the authors demonstrate results on various input transform/label mapping techniques, in terms of backbones only ResNet18 is used, and only on CIFAR-10/100 and GTSRB datasets, which consist of small images, which is qualitatively different from practical image recognition tasks.

**Questions:**

1. The authors could clarify more on how their theoretical work on guaranteed adversarial robustness risk upper bound affects their method and potentially what are the implications of their proof for this task

2. Would these results generalize to more practical datasets such as ImageNet and recent models such as VisionTransformers, which have very different robustness characteristics than CNNs ?

---

> ### Author Response · Authors · 2024-11-21
> **Reply to Reviewer krkJ (Part 1/3)**
>
> Thank you so much for your valuable comments! Our point-by-point responses to the reviewer's mentioned questions (Q) and weaknesses (W) are provided as follows.
>
> **W1: Limited novelty, the main component of the method is using adversarially pre-trained backbones, and using adversarial training during visual re-programming, which is a direct application of prior work.**
>
> We would like to emphasize that our primary goal is to explore the performance of visual reprogramming (VR) methods in adversarial scenarios, providing an important theoretical framework and experimental validation, rather than proposing a new vanilla VR method or a adversarial training technique. As VR methods leverage the potential of pre-trained model capabilities, an increasing number of VR methods have been proposed. However, these methods lack investigations focused on security, which severely hinders their application in security-sensitive domains. Adversarial robustness is a critical metric in security-critical fields, and adversarial training is a commonly used and effective approach for enhancing adversarial robustness. Therefore, in our research, we incorporate widely used adversarial training methods (i.e., PGD and TRADES) into our discussions. Furthermore, to explore the combination of different VR methods in adversarial scenarios, we include commonly used VR methods (i.e., Pad, FULL, RandLM, IterLM, and FreqLM) in our discussions.
>
> As a fundamental research, our work theoretically and empirically demonstrates that the robustness of VR is influenced by various factors, such as the robustness of pre-trained models, the use of adversarial training, and different input transformations and output label mappings. These findings can serve as baseline results and help provide reliable insights for future research.
>
> In our humble opinion, **proposing entirely new methods is not the sole measure of contribution; our work is the first to investigate adversarial robustness in VR, providing theoretical analysis and experimental validation, which we believe represents a foundational and meaningful step in this area.** There are many such foundational works, such as: AutoVP [1], which designs a framework to automate visual prompting for downstream image classification tasks; ARD [2], which combines knowledge distillation with adversarial training, motivates subsequent research.
>
> [1] Hsi-Ai Tsao, Lei Hsiung, Pin-Yu Chen, Sijia Liu, and Tsung-Yi Ho. Autovp: An automated visual prompting framework and benchmark. In ICLR, 2024.
>
> [2]  Goldblum, Micah and Fowl, Liam and Feizi, Soheil and Goldstein, Tom. Adversarially Robust Distillation. In AAAI, 2020.
>
> **W2: The theoretical portion of the paper tries to tackle an interesting question, however the relevance to the method is limited ? The method itself is very empirical and intuitive, and the theory seems like an afterthought.**
>
> Our method is derived from our theoretical framework, and our experimental results validate our theory. They are intrinsically related, and the theory was not introduced subsequently. In Section 5.2, we have added more detailed discussions based on both experimental and theoretical results, including our reasons for using adversarially pre-trained models as backbones and incorporating adversarial examples from the target domain into the reprogramming, highlighted in blue.
>
> Specifically, the first term of the upper bound in Theorem 1 indicates that a pre-trained model with robustness in the source domain can enhance the robustness performance after reprogramming. Therefore, we employed adversarially pre-trained models as the backbone in Strategies 3 and 4, which performed better than Strategy 1, thereby validating the rationale of our theory. Furthermore, the second term of the upper bound in Theorem 1 suggests that the robustness of visual reprogramming is influenced by $f_{\mathrm{in}}$, $f_{\mathrm{out}}$ and and the discrepancy between the source and target domains. Consequently, we utilized various $f_{\mathrm{in}}$ and $f_{\mathrm{out}}$ methods in our experiments. In most cases, SMM outperforms other $f_{\mathrm{in}}$ methods, while FullyLM surpasses other $f_{\mathrm{out}}$ methods. This is because these methods possess more complex function spaces, making it easier to bridge the gap between the source and target domains. These findings are consistent with our theoretical analysis. Additionally, the second term is related to adversarial margin disparity (Equation 12), which implies that incorporating adversarial training into the reprogramming process can mitigate risks, as adversarial margin disparity represents the distance in the adversarial space. Therefore, in Strategies 2 and 4, we employed commonly used adversarial training methods. Our experimental results confirm this, demonstrating that adversarial training consistently enhances the robustness of visual reprogramming.
>
> Continue by the next post.

---

> ### Author Response · Authors · 2024-11-21
> **Reply to Reviewer krkJ (Part 2/3)**
>
> **W3: Though the authors demonstrate results on various input transform/label mapping techniques, in terms of backbones only ResNet18 is used, and only on CIFAR-10/100 and GTSRB datasets, which consist of small images, which is qualitatively different from practical image recognition tasks.**
>
> Following your suggestions, we have added some experiments involving larger images and more general scenarios (Flowers102, OxfordPets, SUN397, and Food101). Additionally, we have included experiments with a variety of different model architectures (ViT-S+ConvStem and XCIT-S12). These experiments further validate our findings and support our theoretical framework. We will provide more comprehensive experiments in the subsequent version of the manuscript.
>
> Table 1: The performance comparison of VR with XCIT-S12 (A.T.) on different datasets, and PGD-20 is used for adversarial testing. Using PGD-10 ($\epsilon=4/255$) as adversarial training. The results are presented in the format of robustness (clean accuracy). The best robustness is highlighted in bold.
>
> |    In / Out    |    Flowers102     |    OxfordPets     |      SUN397       |      Food101      |
> | :------------: | :---------------: | :---------------: | :---------------: | :---------------: |
> | Pad / FullyLM  |   37.99 (68.21)   |   36.61 (77.85)   |   21.67 (51.03)   |   21.47 (45.79)   |
> | Full / FullyLM | **40.88 (74.77)** | **46.81 (86.44)** | **23.99 (58.09)** | **23.25 (52.77)** |
>
> Table 2: The performance of VR with XCIT-S12 (A.T.) under natural training (N.T.) on different datasets, and PGD-20 is used for adversarial testing. The results are presented in the format of robustness (clean accuracy). The best robustness is highlighted in bold.
>
> |    In / Out    |    Flowers102     |    OxfordPets     |      SUN397      |     Food101      |
> | :------------: | :---------------: | :---------------: | :--------------: | :--------------: |
> | Pad / FullyLM  | **29.47 (65.66)** |   28.35 (77.29)   | **7.80 (48.89)** | **9.81 (50.10)** |
> | Full / FullyLM |   28.89 (77.31)   | **38.03 (85.88)** |   7.23 (54.96)   |   9.32 (57.74)   |
>
> Table 3: The performance comparison of VR with XCIT-S12 (N.T.) on different datasets, and PGD-20 is used for adversarial testing. Using PGD-10 ($\epsilon=4/255$) as adversarial training. The results are presented in the format of robustness (clean accuracy). The best robustness is highlighted in bold.
>
> |    In / Out    |    Flowers102    |   OxfordPets    |      SUN397      |     Food101     |
> | :------------: | :--------------: | :-------------: | :--------------: | :-------------: |
> | Pad / FullyLM  | **0.00 (15.77)** | **2.48 (5.55)** |   2.15 (3.01)    |   0.99 (1.00)   |
> | Full / FullyLM | **0.00 (13.11)** |  2.04 (20.76)   | **6.76 (22.17)** | **3.84 (8.85)** |
>
> Table 4: The performance comparison of VR with ViT-S+ConvStem (A.T.) on different datasets, and PGD-20 is used for adversarial testing. Using PGD-10 ($\epsilon=4/255$) as adversarial training. The results are presented in the format of robustness (clean accuracy). The best robustness is highlighted in bold.
>
> |    In / Out    |    Flowers102     |    OxfordPets     |      SUN397       |      Food101      |
> | :------------: | :---------------: | :---------------: | :---------------: | :---------------: |
> | Pad / FullyLM  |   39.47 (63.48)   |   39.82 (76.45)   |   24.84 (53.83)   |   21.09 (45.30)   |
> | Full / FullyLM | **44.28 (71.68)** | **49.41 (82.95)** | **25.84 (57.57)** | **23.36 (50.38)** |
>
> Table 5: The performance of VR with ViT-S+ConvStem (A.T.) under natural training (N.T.) on different datasets, and PGD-20 is used for adversarial testing. The results are presented in the format of robustness (clean accuracy). The best robustness is highlighted in bold.
>
> |    In / Out    |    Flowers102     |    OxfordPets     |      SUN397       |      Food101      |
> | :------------: | :---------------: | :---------------: | :---------------: | :---------------: |
> | Pad / FullyLM  |   34.64 (62.84)   |   35.54 (77.01)   | **11.41 (50.88)** | **11.89 (48.40)** |
> | Full / FullyLM | **39.57 (72.85)** | **43.08 (82.61)** |   10.84 (54.38)   |   11.10 (53.36)   |
>
> Continue by the next post.

---

> > ### Comment · Reviewer_krkJ · 2024-11-25
> > **Interesting results**
> >
> > These additional results are interesting. If I am understanding them correctly, it seems if I compare Table 1 to Table 5, it seems like the effect of Adversarial Training during VR is limited ? Most of the robustness seems to come from using the A.T. pretrained backbone ?

---

> > > ### Author Response · Authors · 2024-11-25
> > > **Thank you for your feedback!**
> > >
> > > We appreciate receiving your response!
> > >
> > > We agree with the your perspective that leveraging adversarially pretrained models can significantly enhance robustness (as demonstrated in Strategy 3 and Strategy 4). Furthermore, adversarial training during VR can further improve robustness, although the improvement may be relatively limited. For instance, in the part (2/3) of our reply, as shown in Table 1 (Strategy 4) compared with Table 2 (Strategy 3) (noting that the pretrained models in Table 5 differ from those in Table 1), adversarial training results in approximately a 10% improvement across all datasets. Similarly, when comparing Table 2 (Strategy 3) to Table 3 (Strategy 2), adversarially pretrained models achieve a substantial improvement of around 30% on Flowers102 and OxfordPets compared to non-adversarially pretrained models. Due to limitations in table formatting of OpenReview, similar findings can be observed in the experimental results (see Table 1, Table 2, Table 3, and Figure 3) presented in our revised manuscript (you can re-download it in OpenReview). We believe that the limited robustness gains from adversarial training may be due to the finite number of trainable parameters in VR. As shown in the Tables and Figures of the revised manuscript, when using VR methods with larger parameter spaces such as SMM and FullyLM, the enhancement of robustness (Strategy 4) through adversarial training is more pronounced. Therefore, adversarial training remains a crucial technique for improving robustness.
> > >
> > > Nonetheless, these observations are not absolute, as adversarial training does not always significantly enhance robustness (e.g., in Flowers102 and OxfordPets as shown in Table 1 (Strategy 4), Table 2 (Strategy 3), and Table 3 (Strategy 2) in the part (2/3) of our reply); similarly, using adversarially pretrained models does not always lead to significant robustness improvements (e.g., in SUN397 and Food101 as shown in Table 1 (Strategy 4), Table 2 (Strategy 3), and Table 3 (Strategy 2) in the part (2/3) of our reply). This aligns with our theoretical findings. In our work, we theoretically and experimentally demonstrate that the robustness of VR is influenced by multiple factors: the robustness of the pretrained model, the use of adversarial training, and different input transformations and output label mappings.
> > >
> > > Therefore, we believe that researching adversarial training methods compatible with the VR framework, as well as designing better $f_\text{in}$ and $f_\text{out}$ methods, are important challenges in endowing VR with robustness. We are confident that our theoretically guaranteed work can inspire further studies to focus on this interesting and important problem.
> > >
> > > We hope that our further response can effectively address your concerns. If you still have any questions about our work, please let us know, and we would be happy to address them for you.

---

> > > > ### Comment · Reviewer_krkJ · 2024-11-25
> > > > **Good response**
> > > >
> > > > Thanks for clarifying this, I would recommend adding some of these additional results and caveats to the paper as well so that the limitations are clear to readers. This would address my key concern.

---

> > > > > ### Author Response · Authors · 2024-11-26
> > > > > **Thanks for your feedback!**
> > > > >
> > > > > Thank you very much for your valuable suggestions.  Based on your feedback, we have included additional experiments with more practical datasets and diverse model architectures in Appendix C. We have also added a discussion in Section 5.2 (highlighted from lines 455 to 463) of the revised manuscript on the impact of various strategies on enhancing robustness of VR. This will help readers better understand the challenges of VR with adversarial robustness and the significance of our work.
> > > > >
> > > > > It is important to note that the challenges presented are not limitations of our study. As the first foundational work on VR with adversarial robustness, our work offers insightful theoretical analysis and experimental validation, laying the groundwork for future research. It is difficult to accomplish all the related tasks, although addressing these challenges is certainly interesting and improtant. We believe that tackling issues such as techniques for $f_{\text{in}}$ and $f_{\text{out}}$ specifically designed for VR with adversarial robustness, and developing adversarial training methods tailored for VR, is an challenge worthy of in-depth investigation and more suitable as a direction for future work. This is also one of the main goals of our work, which is to inspire more researchers to pay attention to this important and interesting problem.
> > > > >
> > > > > If you think we have addressed your concern, please re-evaluate our manuscript. If not, we would greatly appreciate it if you could point out the issues, and we will do our best to respond to you as soon as possible.

---

> > > > > ### Author Response · Authors · 2024-11-29
> > > > > **Looking forward to your response**
> > > > >
> > > > > We deeply appreciate your constructive suggestions and feedback, and sincerely apologize for any inconvenience caused.
> > > > >
> > > > > Following your suggestion, we have incorporated these additional results and caveats into the paper. We would like to clarify that the challenges proposed based on the theoretical and experimental findings in this study do not represent limitations of our research. We believe that tackling these issues represents a research challenge worth exploring in depth, and it is better suited as a direction for future work.
> > > > >
> > > > > We hope our response will address your concerns and better emphasize our contributions. We would be truly grateful If you could consider raising your rating in recognition of these contributions. Regardless of the final rating, we sincerely respect and appreciate your review process and insightful comments, which have been invaluable in improving our work. Due to the limited time for discussion, we look forward to your response.

---

> > > > > ### Author Response · Authors · 2024-12-02
> > > > > **Kind reminder to Reviewer krkJ**
> > > > >
> > > > > Sorry to disturb you. We sincerely appreciate your valuable comments and understand that you may be too busy to review our rebuttal. However, since there is only about one day left for discussion, we would greatly appreciate it if you could kindly take a moment to review our response and provide your feedback.
> > > > >
> > > > > We genuinely hope that our response will address your concerns and look forward to your response.

---

> > > > > > ### Comment · Reviewer_krkJ · 2024-12-02
> > > > > >
> > > > > > I appreciate the improvements in the paper. I believe the changes merit an increase in rating to marginally/borderline above acceptance threshold.

---

> > > > > > > ### Author Response · Authors · 2024-12-03
> > > > > > > **Thank you!**
> > > > > > >
> > > > > > > Thank you so much for your support to our work! We really appreciate your valuable comments and constructive suggestions, which definitely have helped us to improve the quality of our paper!

---

> ### Author Response · Authors · 2024-11-21
> **Reply to Reviewer krkJ (Part 3/3)**
>
> **Q1: The authors could clarify more on how their theoretical work on guaranteed adversarial robustness risk upper bound affects their method and potentially what are the implications of their proof for this task.**
>
> Specifically, the first term of the upper bound in Theorem 1 suggests that a source domain pre-trained model with inherent robustness can improve robustness performance post-reprogramming. Consequently, we utilized adversarially pre-trained models as the foundation in Strategies 3 and 4. These strategies outperformed Strategy 1, thereby supporting the theoretical basis of our approach. Additionally, the second term of the upper bound in Theorem 1 implies that the robustness of visual reprogramming is affected by $f_{\mathrm{in}}$, $f_{\mathrm{out}}$ and the disparity between the source and target domains. As a result, we implemented various $f_{\mathrm{in}}$ and $f_{\mathrm{out}}$ methods in our experiments. In most instances, SMM outperformed other $f_{\mathrm{in}}$ methods, while FullyLM exceeded other $f_{\mathrm{out}}$ methods. This is attributed to the more complex function spaces of these methods, which facilitate bridging the gap between source and target domains. These results align with our theoretical analysis. Furthermore, the second term relates to adversarial margin disparity, indicating that integrating adversarial training into the reprogramming process can reduce risks, as adversarial margin disparity reflects the distance in the adversarial space. Therefore, in Strategies 2 and 4, we employed widely-used adversarial training techniques. Our experimental findings corroborate this, showing that adversarial training consistently enhances the robustness of visual reprogramming.
>
>
> Our theory demonstrates that the robustness of VR is influenced by various factors (i.e., the robustness of pre-trained models, the use of adversarial training, and different input transformations and output label mappings). These insights can inspire the design of pre-trained models specifically tailored for adversarially robust VR (the first term in theorem), methods to evaluate and optimize the robustness distance between two domains (the second term in theorem), techniques for $f_{\mathrm{in}}$ and $f_{\mathrm{out}}$ specifically designed for adversarially robust VR (the second term in theorem), adversarial training methods specifically designed for adversarially robust VR (the second term in theorem) and optimization strategies targeting the third term of our theoretical framework.
>
> **Q2: Would these results generalize to more practical datasets such as ImageNet and recent models such as VisionTransformers, which have very different robustness characteristics than CNNs ?**
>
> We greatly appreciate your suggestions. Based on your suggestions, we have included some experiments involving more practical datasets and updated model architectures as shown in the reply of W3. We will provide more comprehensive experimental results in future versions of our manuscript. These experimental results also demonstrate the effectiveness of our strategy and validate our theory.
>
> We hope that our responses can satisfactorily address your concerns. Thank you again for your time to provide us with valuable feedback.

---

> ### Author Response · Authors · 2024-11-24
> **Looking forward to your response**
>
> We greatly appreciate your involvement in our work. Your insightful feedback has helped us enhance our work. We have addressed your concerns and suggestions in the rebuttal, and we would be much appreciated if you could kindly check our rebuttal. Please let us know if you have any other questions, and we would be delighted to continue the discussion with you. We are grateful for your efforts in improving our work.

---

### Official Review · Reviewer_xfee · 2024-11-04

**Soundness:** 2
**Presentation:** 3
**Contribution:** 2
**Rating:** 5
**Confidence:** 2

**Summary:**

This paper focuses on the adversarial robustness of visual reprogramming (VR) against adversarial attacks. To this end, it first empirically finds that reprogramming pre-trained models with adversarial robustness and incorporating adversarial samples from the target task during reprogramming can both improve the adversarial robustness of reprogrammed models. Furthermore, it also presents a theoretically guaranteed adversarial robustness risk upper bound for VR. Experiments on several benchmarks validate the effectiveness of the proposed method.

**Strengths:**

1. The proposed method is interesting. The paper is well-written, and easy to understand.
2. The authors introduced an upper bound for the adversarial robustness risk in visual reprogramming, theoretically proving the effectiveness of the proposed method.
3. Extensive comparative experiments were conducted on multiple datasets, showcasing the performance of the proposed method across different attack iteration numbers and perturbation sizes.

**Weaknesses:**

1. The technical novelty of this paper is limited. The modules for visual programming, such as input transformation and label mapping, are similar to (Chen, 2022; Cai et al. 2024). And those modules for adversarial robustness also have been proposed by previous literature (Madry et al. 2018; Zhang et al., 2019a)
2. I'm a little confused about the task setting, especially in the availability of adversarially pre-trained models. For the source domain, it may not be practical to have both a pre-trained model and an adversarially pre-trained model. Obviously, we can’t temporarily train a source model via adversarial training. if we can, why don't we train a target model directly with clean and adversarial data, because both datasets are also available?
3. Experiments in Sec. 5 are more like ablation studies, to discuss the effectiveness of different designs for the proposed framework. However, it lacks comparisons with kinds of baselines or SOTA methods. Moreover, it would be more significant to evaluate the proposed method with more complex models on more large datasets, because that is the point of visual reprogramming to reduce the training and storage costs for target domains.
4. In Eq. 10, $\Phi_{\rho}$ has two input variables (i.e., $x$ and $y$), but in the definition of Eq. 11, it only has one.

**Questions:**

Please see the Weaknesses for details.

---

> ### Author Response · Authors · 2024-11-21
> **Reply to Reviewer xfee (Part 1/3)**
>
> We sincerely thank you for your thorough review and constructive comments on our manuscript. We appreciate your recognition of the clarity and proposed method. Your feedback has provided us with valuable insights that will help us further improve our work. Next, we will address your questions (Q) and weaknesses (W) in detail below.
>
> **W1: The technical novelty of this paper is limited. The modules for visual programming, such as input transformation and label mapping, are similar to (Chen, 2022; Cai et al. 2024). And those modules for adversarial robustness also have been proposed by previous literature (Madry et al. 2018; Zhang et al., 2019a)**
>
> We would like to emphasize that the primary goal of our work is to investigate the performance of Visual Reprogramming (VR) in adversarial scenarios, providing both a solid theoretical framework and experimental validation. While many VR methods have been proposed in the existing literature (e.g., [1]), there is a lack of research targeting safety-critical tasks, where adversarial robustness is particularly important. In these contexts, adversarial training (e.g., [2], [3]) is the most effective and widely adopted strategies to enhance model robustness.
>
> Our work serves as a foundational study, theoretically and experimentally demonstrating that the robustness of VR is influenced by multiple factors: the robustness of the pretrained model, the use of adversarial training, and different input transformations and output label mappings. These findings can act as baseline results, helping to reduce redundancy in future research while providing reliable insights for subsequent innovations.
>
> In our humble opinion, **proposing entirely new methods is not the sole measure of contribution; our work is the first to investigate adversarial robustness in VR, providing theoretical analysis and experimental validation, which we believe represents a foundational and meaningful step in this area.** There are many such foundational works, such as: AutoVP [4], which designs a framework to automate visual prompting for downstream image classification tasks; ARD [5], which combines knowledge distillation with adversarial training, motivates subsequent research.
>
> Additionally, our work differs significantly from [6]. Their study can be understood as focusing on scenarios where the source and target domains are the same. For example, the experiments in [6] only utilize non-robust pretrained models on CIFAR-10 and leverage VP to add prompts to the inputs, achieving adversarial robustness on CIFAR-10. In contrast, our work focuses on the adversarial robustness of VR, where the source and target domains may differ. For instance, we use pretrained models on ImageNet and apply them to target tasks such as CIFAR-10 or GTSRB. As a result, we not only adopt different input transformations but also employ distinct output label mappings to adapt to the target domain. Furthermore, we evaluate the impact of adversarially pretrained models and non-adversarially pretrained models, which is different from the setting in [6].
>
> [1] Chengyi Cai, Zesheng Ye, Lei Feng, Jianzhong Qi, and Feng Liu. Sample-specific masks for visual reprogramming-based prompting. In ICML, pp. 5383–5408, 2024.
>
> [2] Aleksander Madry, Aleksandar Makelov, Ludwig Schmidt, Dimitris Tsipras, and Adrian Vladu. Towards deep learning models resistant to adversarial attacks. In ICLR, 2018.T
>
> [3] Hongyang Zhang, Yaodong Yu, Jiantao Jiao, Eric Xing, Laurent El Ghaoui, and Michael Jordan. Theoretically principled trade-off between robustness and accuracy. In ICML, pp. 7472–7482,
> 2019a.
>
> [4] Hsi-Ai Tsao, Lei Hsiung, Pin-Yu Chen, Sijia Liu, and Tsung-Yi Ho. Autovp: An automated visual prompting framework and benchmark. In ICLR, 2024.
>
> [5] Goldblum, Micah and Fowl, Liam and Feizi, Soheil and Goldstein, Tom. Adversarially Robust Distillation. In AAAI, 2020.
>
> [6] Aochuan Chen, Peter Lorenz, Yuguang Yao, Pin-Yu Chen, and Sijia Liu. Visual prompting for adversarial robustness. In ICASSP 2023, pp. 1–5, 2023a.
>
> Continue by the next post.

---

> ### Author Response · Authors · 2024-11-21
> **Reply to Reviewer xfee (Part 2/3)**
>
> **W2: I'm a little confused about the task setting, especially in the availability of adversarially pre-trained models. For the source domain, it may not be practical to have both a pre-trained model and an adversarially pre-trained model. Obviously, we can’t temporarily train a source model via adversarial training. if we can, why don't we train a target model directly with clean and adversarial data, because both datasets are also available?**
>
> The purpose of VR is to apply well-trained pretrained models to other tasks without the need to retrain or modify the parameters of the pretrained models from scratch. There are many publicly available adversarially pretrained models with strong adversarial robustness, which can be obtained from platforms like [RobustBench](https://github.com/RobustBench/robustbench).
>
> In practical applications, it is not necessary to simultaneously acquire both non-adversarially pretrained models and adversarially pretrained models. In our experiments, we set up this comparison to explore the impact of these two types of pretrained models on the robustness of VR and to validate the first term in the upper bound of Theorem 1: the influence of the adversarial risk of pretrained models on the source domain on VR robustness. Our results demonstrate that using adversarially pretrained models tends to provide better robustness for VR, which can inspire future research to better leverage the capabilities of adversarially pretrained models.
>
> Moreover, it might be possible to directly train a robust model on the target domain, but this would be computationally inefficient to train a model from scratch. In contrast, as stated above, the purpose of VR is to better utilize existing pretrained models, with only a few learnable parameters. Therefore, VR can significantly reduce the training cost, making it possible to obtain larger-scale models with limited training resources.
>
> **W3: Experiments in Sec. 5 are more like ablation studies, to discuss the effectiveness of different designs for the proposed framework. However, it lacks comparisons with kinds of baselines or SOTA methods. Moreover, it would be more significant to evaluate the proposed method with more complex models on more large datasets, because that is the point of visual reprogramming to reduce the training and storage costs for target domains.**
>
> Our experiments include four strategies in terms of whether to use adversarially pre-trained models and whether to apply adversarial training. Among these, the strategy S1 employing non-adversarially pretrained models and non-adversarial training corresponds to the standard approach adopted in existing VR works, which can be considered as the baseline method. Moreover, because adversarial robustness remains an underexplored aspect in VR, no existing methods are explicitly designed to address this problem. Our work demonstrates that current VR methods are vulnerable to adversarial attacks and that leveraging adversarially pretrained models and adversarial training can enhance the adversarial robustness of VR. We believe this contribution provides both theoretical and empirical groundwork for future research in this area.
>
> In order to further verify the effectiveness of our strategy on more complex models and larger datasets, we conducte additional experiments with ViT and XCiT on larger datasets, including Flowers102, OxfordPets, SUN397, and Food101 according to your suggestions:
>
> Table 1: The performance comparison of VR with XCIT-S12 (A.T.) on different datasets, and PGD-20 is used for adversarial testing. Using PGD-10 ($\epsilon=4/255$) as adversarial training. The results are presented in the format of robustness (clean accuracy). The best robustness is highlighted in bold.
>
> |    In / Out    |    Flowers102     |    OxfordPets     |      SUN397       |      Food101      |
> | :------------: | :---------------: | :---------------: | :---------------: | :---------------: |
> | Pad / FullyLM  |   37.99 (68.21)   |   36.61 (77.85)   |   21.67 (51.03)   |   21.47 (45.79)   |
> | Full / FullyLM | **40.88 (74.77)** | **46.81 (86.44)** | **23.99 (58.09)** | **23.25 (52.77)** |
>
> Table 2: The performance of VR with XCIT-S12 (A.T.) under natural training (N.T.) on different datasets, and PGD-20 is used for adversarial testing. The results are presented in the format of robustness (clean accuracy). The best robustness is highlighted in bold.
>
> |    In / Out    |    Flowers102     |    OxfordPets     |      SUN397      |     Food101      |
> | :------------: | :---------------: | :---------------: | :--------------: | :--------------: |
> | Pad / FullyLM  | **29.47 (65.66)** |   28.35 (77.29)   | **7.80 (48.89)** | **9.81 (50.10)** |
> | Full / FullyLM |   28.89 (77.31)   | **38.03 (85.88)** |   7.23 (54.96)   |   9.32 (57.74)   |
>
> Continue by the next post.

---

> ### Author Response · Authors · 2024-11-21
> **Reply to Reviewer xfee (Part 3/3)**
>
> Table 3: The performance comparison of VR with XCIT-S12 (N.T.) on different datasets, and PGD-20 is used for adversarial testing. Using PGD-10 ($\epsilon=4/255$) as adversarial training. The results are presented in the format of robustness (clean accuracy). The best robustness is highlighted in bold.
>
> |    In / Out    |    Flowers102    |   OxfordPets    |      SUN397      |     Food101     |
> | :------------: | :--------------: | :-------------: | :--------------: | :-------------: |
> | Pad / FullyLM  | **0.00 (15.77)** | **2.48 (5.55)** |   2.15 (3.01)    |   0.99 (1.00)   |
> | Full / FullyLM | **0.00 (13.11)** |  2.04 (20.76)   | **6.76 (22.17)** | **3.84 (8.85)** |
>
> Table 4: The performance comparison of VR with ViT-S+ConvStem (A.T.) on different datasets, and PGD-20 is used for adversarial testing. Using PGD-10 ($\epsilon=4/255$) as adversarial training. The results are presented in the format of robustness (clean accuracy). The best robustness is highlighted in bold.
>
> |    In / Out    |    Flowers102     |    OxfordPets     |      SUN397       |      Food101      |
> | :------------: | :---------------: | :---------------: | :---------------: | :---------------: |
> | Pad / FullyLM  |   39.47 (63.48)   |   39.82 (76.45)   |   24.84 (53.83)   |   21.09 (45.30)   |
> | Full / FullyLM | **44.28 (71.68)** | **49.41 (82.95)** | **25.84 (57.57)** | **23.36 (50.38)** |
>
> Table 5: The performance of VR with ViT-S+ConvStem (A.T.) under natural training (N.T.) on different datasets, and PGD-20 is used for adversarial testing. The results are presented in the format of robustness (clean accuracy). The best robustness is highlighted in bold.
>
> |    In / Out    |    Flowers102     |    OxfordPets     |      SUN397       |      Food101      |
> | :------------: | :---------------: | :---------------: | :---------------: | :---------------: |
> | Pad / FullyLM  |   34.64 (62.84)   |   35.54 (77.01)   | **11.41 (50.88)** | **11.89 (48.40)** |
> | Full / FullyLM | **39.57 (72.85)** | **43.08 (82.61)** |   10.84 (54.38)   |   11.10 (53.36)   |
>
> These experimental results have been added to Appendix C, and We will provide more comprehensive experiments in the subsequent version of the manuscript. These experimental results demonstrate the effectiveness of our strategy on more complex models as well as on larger datasets.
>
> **W4: In Eq. 10, $\Phi_\rho$ has two input variables (i.e., $x$ and $y$), but in the definition of Eq. 11, it only has one.**
>
>  Thank you for your suggestion. There is some ambiguity in the expression of Eq. 10. To provide a clearer expression, we have rewritten Eq. 10 as follows:
>
> $$
> \mathcal{R} _{\mathcal{U}}^{\text{adv}, (\rho)}(f) \triangleq \mathbb{E} _{(\boldsymbol{x}, y)\sim \mathcal{U}}\left[ \max _{\Vert\gamma\Vert\leq\epsilon} \Phi _\rho \left( \rho _f(\boldsymbol{x}+\gamma,y) \right) \right]
> $$
>
> We hope that our responses can satisfactorily address your concerns. Thank you again for your time to provide us with valuable feedback.

---

> ### Author Response · Authors · 2024-11-24
> **Looking forward to your response**
>
> We sincerely appreciate you taking the time to provide constructive feedback on our manuscript. In our rebuttal, we have addressed your concerns and questions, particularly regarding the technical novelty and the task setting.  If you have any further concerns or suggestions, we would be happy to continue the discussion with you! We once again appreciate your efforts in enhancing our work.

---

> > ### Comment · Reviewer_xfee · 2024-11-26
> >
> > I apologize for the late reply and appreciate the authors’ efforts in responding to comments, especially experiments on larger datasets. Based on this, I am willing to upgrade my score to 5.
> >
> > However, my first and third questions are not fully addressed, leaving concerns about the technical novelty. First, both Visual Reprogramming (VR) and Adversarial Training (AT) have been widely studied, thus, the technical contribution of just combining the two as the first work is not enough. Second, no offense to belittle this work, but in my opinion, using adversarially pre-trained models is clearly effective for improving robustness. However, this effectiveness is limited by whether pre-trained models are available from RobustBench. As seen from Tabs. 1 and 2, the robustness is significantly reduced if adversarially pre-trained models are not available. It would be more expected to see the authors' insight and exploration of this limitation, and about how to alleviate it.

---

> > > ### Author Response · Authors · 2024-11-27
> > > **Thank you for your response!**
> > >
> > > Thank you for your careful review and valuable comments on our work. We provide detailed responses to your questions here.
> > >
> > > **Q1: Both Visual Reprogramming (VR) and Adversarial Training (AT) have been widely studied, thus, the technical contribution of just combining the two as the first work is not enough.**
> > >
> > > We agree that Visual Reprogramming (VR) and Adversarial Training (AT) have been widely studied; however, their synergistic effects remain underexplored.
> > >
> > > As Reviewer R93c acknowledged our contribution:"The article not only experimentally verifies that using adversarially pre-trained models and adversarial samples can improve the adversarial robustness of VR models but also proposes a theoretical guarantee of an adversarial robustness risk upper bound, providing a theoretical foundation for future research." Our work theoretically and experimentally investigates the effectiveness of various strategies and VR methods in enhancing adversarial robustness. It demonstrates that the robustness of VR is influenced by multiple factors, including the robustness of the pre-trained model, the use of adversarial training, and different input transformations and output label mappings. These findings introduce two entirely new challenges, which also represent significant contributions of our work:
> > >
> > > - **Compatibility Between Adversarial Training Methods and VR Frameworks**: In VR, the trainable components are limited to the input transformation $f_\text{in}$ and the output label mapping $f_\text{out}$. This restriction poses a distinct challenge in designing adversarial training algorithms, which differ significantly from existing adversarial training research.
> > > - **Design of  $f_\text{in}$ and $f_\text{out}$**: Our theoretical and experimental results reveal that different choices of $f_\text{in}$ and $f_\text{out}$ significantly impact adversarial robustness. However, existing VR methods do not consider adversarial robustness in their design. For example, existing VR research has not structurally considered the design of $f_\text{in}$ and $f_\text{out}$ to be inherently robust, nor has it addressed their compatibility with adversarial training methods. Therefore, designing $f_\text{in}$ and $f_\text{out}$ with inherent consideration for adversarial robustness represents a unique challenge in advancing the robustness of VR.
> > >
> > > As a foundational work (especially for a conference paper), it is difficult to address all related tasks comprehensively, although solving these challenges is undoubtedly both interesting and important. We believe that tackling these issues represents a research challenge worth exploring in depth, and it is better suited as a direction for future work. This is also one of the main goals of our paper: to inspire more researchers to focus on this interesting and important problem.
> > >
> > > **Q2: Using adversarially pre-trained models is clearly effective for improving robustness. However, this effectiveness is limited by whether pre-trained models are available from RobustBench. As seen from Tabs. 1 and 2, the robustness is significantly reduced if adversarially pre-trained models are not available.**
> > >
> > > Using non-adversarially pre-trained models indeed results in lower adversarial robustness, which is consistent with our theoretical analysis. We understand the concern regarding the availability of adversarially pre-trained models, but we would like to clarify that one of the advantages of VR is its ability to apply the same pre-trained model to different target tasks. Therefore, we do not need to select different adversarially pre-trained models for different tasks; one adversarially pre-trained model is sufficient, which significantly reduces the need for acquiring the adversarially pre-trained models.
> > >
> > > Additionally, we would like to point out that obtaining an adversarially pre-trained model is not a difficult task nowadays. For instance, RobustBench (https://github.com/RobustBench/robustbench) offers a wide range of adversarially pre-trained models with strong robustness, designed to be easy to use for any downstream application. These models are uploaded by researchers and continue to be updated. Besides platforms like RobustBench, which aggregate adversarially pre-trained models, there is an increasing number of works in the adversarial training domain, making it easy to access their adversarially pre-trained models from publicly available code repositories.
> > >
> > > We hope that our response satisfactorily address your questions. Thank you again for supporting our work and taking the time to provide us with valuable feedback.

---

> > > ### Author Response · Authors · 2024-11-29
> > > **Looking forward to your response**
> > >
> > > We deeply appreciate your constructive suggestions and feedback, and sincerely apologize for any inconvenience caused.
> > >
> > > We have reiterated the theoretical and experimental findings of our work and elaborated on the two novel challenges we proposed in the adversarial robustness of visual reprogramming (VR) to more clearly emphasize our contributions. Furthermore, we highlighted  that a single adversarially pre-trained model can be applied across various target tasks based on the advantages of VR. We also provide some examples of obtaining adversarially pretrained models to illustrate how accessible obtaining an adversarially pre-trained model has become in practice.
> > >
> > > We hope our response will address your concerns and better emphasize our contributions. We would be truly grateful If you could consider raising your rating in recognition of these contributions. Regardless of the final rating, we sincerely respect and appreciate your review process and insightful comments, which have been invaluable in improving our work. Due to the limited time for discussion, we look forward to your response.

---

> > > ### Author Response · Authors · 2024-12-02
> > > **Kind reminder to Reviewer xfee**
> > >
> > > Sorry to disturb you. We sincerely appreciate your valuable comments and understand that you may be too busy to review our rebuttal. However, since there is only about one day left for discussion, we would greatly appreciate it if you could kindly take a moment to review our response and provide your feedback.
> > >
> > > We genuinely hope that our response will address your concerns and look forward to your response.

---

> > > > ### Comment · Reviewer_xfee · 2024-12-02
> > > >
> > > > Thanks for the authors' response and clarification. However, I still have concerns on these two points. In this case, I will maintain my current score and defer to the other reviewers for their thoughts on this work.

---

> > > > > ### Author Response · Authors · 2024-12-02
> > > > > **Thank you for your review and feedback!**
> > > > >
> > > > > Thank you very much for the time and effort you have invested to reviewing our work. We sincerely appreciate your comprehensive review and thoughtful feedback, which have motivated us to further improve our study. Regarding these two points, we would like to have the following discussion:
> > > > >
> > > > > - As supported by Reviewer R93c in Strengths, our pioneering exploration into adversarial robustness in VR, along with our proposed solutions and theoretical guarantees on the upper bound of adversarial robustness risk, fills a gap in this field and provides a theoretical foundation for future research.
> > > > > - We believe that obtaining an adversarially pre-trained model is easy nowadays. In fact, RobustBench (https://github.com/RobustBench/robustbench) hosts a wide range of adversarially pre-trained models developed since 2019. Moreover, RobustBench serves merely as a tool for integrating and facilitating the use of such models; researchers can still access the source code and models provided directly by the original authors, bypassing RobustBench if needed. Our findings demonstrate that using adversarially pre-trained models enhances robustness, and we advocate for shifting the focus toward designing better adversarial training methods, as well as improving $f_\text{in}$ and $f_\text{out}$ to maximize the potential of these adversarially pre-trained models.
> > > > >
> > > > > We are deeply grateful for the opportunity to exchange ideas with you and greatly value your insights, which have helped us improve our work. We believe our research can advance the development and application of VR in the field of security and achieve meaningful progress in the future.

---

### Official Review · Reviewer_1Jug · 2024-11-09

**Soundness:** 2
**Presentation:** 3
**Contribution:** 2
**Rating:** 6
**Confidence:** 4

**Summary:**

This paper delves into the adversarial robustness of visual reprogramming. It first highlights the adversarial vulnerability of existing VR methods and proposes to alleviate this issue with two strategies, adopting adversarial trained models and adversarial training in VR. A theoretical analysis is also provided to support the practices.

**Strengths:**

1. The adversarial robustness of visual reprogramming is a critical issue.
2. The proposed methods significantly improve the adversarial robustness.
3. The theoretical upper bound of adversarial risks is straightforward yet valuable, validating the empricial findings.
4. Multiple strategies of VR are considered in experiments.

**Weaknesses:**

1. The two proposed strategies are mature techniques in the field. Technically, it's simply an extension to visual reprogramming and there are no surprising results. Meanwhile, as the number of learnable parameters in visual reprogramming is limited, adversarial training may not be an ideal solution to improve the robustness.
2. Although the theoretical analysis correponds to the proposed strategies, the AMDD-VRA part, which takes up the most context, is not reflected or applied in practice. This term seems to be the most important factor in VR, which is neither taken into consideration in methods nor discussed based on experimental results. For instance, the performance-robustness trade-off on GT-SRB is more noticeable than that on CIFAR. Does it has anything to do with the gap between the source domain and the target domain?
3. There can be more discussions on the recent advancements in Visual Reprogramming and Visual Prompting. For instance, [1] has studied VP with adversarial robustness, what's the difference between your work and theirs. Also, [2] has extended VP to visual foundation models, is your work applicable to more significant scenarios like MLLMs?

Minor: There are some typos in submission, e.g., 'iss' in line 191.

[1] Chen, Aochuan, et al. "Visual prompting for adversarial robustness." ICASSP 2023-2023 IEEE International Conference on Acoustics, Speech and Signal Processing (ICASSP). IEEE, 2023.
[2] Zhang, Yichi, et al. "Exploring the Transferability of Visual Prompting for Multimodal Large Language Models." Proceedings of the IEEE/CVF Conference on Computer Vision and Pattern Recognition. 2024.

**Questions:**

1. Why $f_{out}$ is concatenated to $f_\mathcal{S}$ in the last term of the equation in Definition 1 ? Is there any intuitive explanation on why the source domain $\mathcal{S}$ should also be mapped to the $\mathbb{R}^\mathcal{T}$ ?

I will raise my rating if the authors can properly addressed my concerns.

---

> ### Author Response · Authors · 2024-11-21
> **Reply to Reviewer 1Jug (Part 1/3)**
>
> We sincerely appreciate your time and valuable feedback on our manuscript. Your recognition of the effectiveness and importance of our theoretically guaranteed, simple yet effective research is truly inspiring. The minor issues you pointed out will be carefully addressed in the revised version of the manuscript. Our point-by-point responses to the reviewer's mentioned questions (Q) and weaknesses (W) are provided as follows.
>
> **W1: The two proposed strategies are mature techniques in the field. Technically, it's simply an extension to visual reprogramming and there are no surprising results. Meanwhile, as the number of learnable parameters in visual reprogramming is limited, adversarial training may not be an ideal solution to improve the robustness.**
>
> We agree with the reviewer that the proposed strategies are based on well-established techniques. However, we would like to emphasize that our main contribution is to provide a pioneer study of adversarial robustness for visual reprogramming (VR). The significance of VR lies in applying existing pre-trained models to new tasks while reducing the computational costs. However, the current VR field lacks research on scenarios where the target tasks are security-critical. Therefore, to address this gap, our work serves as a foundational study, theoretically and experimentally demonstrating that the robustness of VR is influenced by multiple factors: the robustness of the pretrained model, the use of adversarial training, and different input transformations and output label mappings. We believe that this comprehensive framework can guide future research to improve the robustness of VR by combining more robust pre-trained models with stronger adversarial training specifically tailored for VR.
>
> In our humble opinion, **proposing entirely new methods is not the sole measure of contribution; our work is the first to investigate adversarial robustness in VR, providing theoretical analysis and experimental validation, which we believe represents a foundational and meaningful step in this area.** There are many such foundational works, such as: AutoVP [1], which designs a framework to automate visual prompting for downstream image classification tasks; ARD [2], which combines knowledge distillation with adversarial training, motivates subsequent research.
>
> Additionally, we agree that the number of learnable parameters in visual reprogramming is limited; however, we would like to emphasize that adversarial training remains a crucial technique for enhancing model robustness in adversarial machine learning. To defend against adversarial attacks, as listed in Section IV of the review [3], most adversarial defense approaches are based on adversarial training or propose improved methods for adversarial training. Moreover, our experimental results demonstrate that adversarial training can indeed effectively enhance adversarial robustness in visual reprogramming. Therefore, we believe that adversarial training remains an important method to improve the robustness of models in adversarial settings, even when the number of learnable parameters is constrained. Furthermore, we also hope that our proposed comprehensive theoretical framework will inspire further research to develop methods for enhancing the adversarial robustness of visual reprogramming with a limited number of learnable parameters.
>
> [1] Hsi-Ai Tsao, Lei Hsiung, Pin-Yu Chen, Sijia Liu, and Tsung-Yi Ho. Autovp: An automated visual prompting framework and benchmark. In ICLR, 2024.
>
> [2]  Goldblum, Micah and Fowl, Liam and Feizi, Soheil and Goldstein, Tom. Adversarially Robust Distillation. In AAAI, 2020.
>
> [3] Baoyuan Wu and Shaokui Wei and Mingli Zhu and Meixi Zheng and Zihao Zhu and Mingda Zhang and Hongrui Chen and Danni Yuan and Li Liu and Qingshan Liu. Defenses in Adversarial Machine Learning: A Survey. arXiv, 2023.
>
> Continue by the next post.

---

> ### Author Response · Authors · 2024-11-21
> **Reply to Reviewer 1Jug (Part 2/3)**
>
> **W2: Although the theoretical analysis correponds to the proposed strategies, the AMDD-VRA part, which takes up the most context, is not reflected or applied in practice. This term seems to be the most important factor in VR, which is neither taken into consideration in methods nor discussed based on experimental results. For instance, the performance-robustness trade-off on GT-SRB is more noticeable than that on CIFAR. Does it has anything to do with the gap between the source domain and the target domain?**
>
> We have added a more detailed discussion based on experimental and theoretical results in Section 5.2, highlighted in blue. Our experimental results indicate that visual reprogramming with adversarially pre-trained models yields greater robustness compared to using naturally pre-trained models. This aligns with the first term of the upper bound in Theorem 1, which suggests that pre-trained models with differing levels of robustness exhibit distinct adversarial risks in the source domain. Consequently, employing a more robust pre-trained model may lead to a more robust visual reprogramming model.
>
> In addition, the second term of the upper bound in Theorem 1 suggests that the robustness of visual reprogramming is influenced by $f_\text{in}$, $f_\text{out}$, and the disparity between the source and target domains. For instance, in most cases, SMM outperforms other $f_\text{in}$ methods, and FullyLM outperforms other $f_\text{out}$ methods. These findings are consistent with our theoretical analysis, indicating that designing better $f_\text{in}$ and $f_\text{out}$ can lead to improved robustness. Furthermore, the second term of the upper bound in Theorem 1 is related to adversarial margin disparity (Eq. 12). This implies that incorporating adversarial training during reprogramming can reduce the risk. Our experimental results confirm this, as adversarial training consistently enhances the robustness of visual reprogramming.
>
> The second term of the upper bound in Theorem 1 also indicates that the robustness of visual reprogramming is affected by the discrepancy between the source and target domains. As the reviewer noted, the performance-robustness trade-off on GTSRB is more noticeable than that on CIFAR. This discrepancy can likely be attributed to the fact that the models used in our experiments were pretrained on ImageNet. ImageNet and CIFAR both involve natural object classification tasks, whereas GTSRB focuses on traffic signs, which are more specific and differ significantly in style and context from the complex natural backgrounds in ImageNet. Since the purpose of $f_\text{in}$ is to serve as an input transformer, designing more powerful $f_\text{in}$ (e.g., SMM) may further reduce the gap between the source and target domains. For example, as shown in Figure 3, the performance gap between Full/FullyLM and SMM/FullyLM is relatively small on the CIFAR-10 dataset, whereas on GTSRB, the robustness of SMM/FullyLM is significantly higher than that of Full/FullyLM.
>
> Continue by the next post.

---

> ### Author Response · Authors · 2024-11-21
> **Reply to Reviewer 1Jug (Part 3/3)**
>
> **W3: There can be more discussions on the recent advancements in Visual Reprogramming and Visual Prompting. For instance, [1] has studied VP with adversarial robustness, what's the difference between your work and theirs. Also, [2] has extended VP to visual foundation models, is your work applicable to more significant scenarios like MLLMs?**
>
> The task settings of Visual Reprogramming (VR) and Visual Prompting (VP) indeed differ significantly. In VR, the source domain and target domain can belong to entirely different domains, as demonstrated in [IBM/model-reprogramming](https://github.com/IBM/model-reprogramming), where the source and target domains may even span different modalities (e.g., images, text, or audio). This allows for the flexible selection of various source domains for VR operations. Furthermore, Section 2.2 of [3] highlights that existing VP methods differ in terms of the placement and functionality of prompts, whereas VR offers a model-agnostic prompting technique. This involves adding trainable noise to input images before forward propagation, which neither alters their visual essence nor relies on specific model architectures.
>
> The differences between our work and [1] align with the above description. In [1], the pretrained model and the dataset used to generate prompts belong to the same dataset (i.e., CIFAR-10). In contrast, in our work, the source domain dataset for the pretrained model and the target dataset for VR are distinct. For instance, we use various models pretrained on ImageNet while employing CIFAR-10 as the target dataset. Additionally, [1] only considers non-robust pretrained models, whereas we analyze the impact of robust and non-robust pretrained models from both theoretical and experimental perspectives.
>
> Due to the differences between the source and target domains in our work, we also utilize distinct visual input transformations and introduce output label mapping, which is not employed in [1], to perform experiments with different combinations. As a foundational study, our work explores the potential of endowing VR with adversarial robustness from both theoretical and experimental angles.
>
> Finally, our research holds significant potential for practical applications, particularly in security-critical MLLMs. Previous studies, such as those cited in [IBM/model-reprogramming](https://github.com/IBM/model-reprogramming), have already demonstrated the powerful potential of reprogramming techniques across different tasks and modalities. For example, when applying general-purpose MLLMs to financial transaction scenarios, the lack of task-specific training may result in elevated security risks. Our work can provide a viable direction to address this challenge.
>
> [1] Aochuan Chen, Peter Lorenz, Yuguang Yao, Pin-Yu Chen, and Sijia Liu. Visual prompting for adversarial robustness. In ICASSP 2023, pp. 1–5, 2023a.
>
> [2] Zhang, Yichi, et al. "Exploring the Transferability of Visual Prompting for Multimodal Large Language Models." Proceedings of the IEEE/CVF Conference on Computer Vision and Pattern Recognition. 2024.
>
> [3] Chengyi Cai, Zesheng Ye, Lei Feng, Jianzhong Qi, and Feng Liu. Sample-specific masks for visual reprogramming-based prompting. In ICML, pp. 5383–5408, 2024.
>
> **W4: Minor: There are some typos in submission, e.g., 'iss' in line 191.**
>
> Thank you for pointing out these issues, which will be very helpful for us to improve our paper. We have rechecked our manuscript and corrected the typos and clarity issues in the revised manuscript.
>
> **Q1: Why $f_\text{out}$ is concatenated to $f_\mathcal{S}$ in the last term of the equation in Definition 1? Is there any intuitive explanation on why the source domain $\mathcal{S}$ should also be mapped to the $\mathbb{R}^{\mathcal{T}}$?**
>
> AMDD-VRA (Definition 1) measures the adversarial margin disparity between the source and target domains within hypothesis spaces $\mathcal{F} _{\mathcal{S}}$ . Since the label spaces $\mathcal{Y}$ of the source and target domains differ, we add the label mapping method to map the source domain label space $\mathcal{Y} _{\mathcal{S}}$ to the target domain $\mathcal{Y} _{\mathcal{T}}$. These methods typically consider only certain labels from the source domain (e.g., FullyLM uses a one-to-one mapping based on frequency), so the labels not considered do not impact VR. In the last term of the equation in Definition 1, $f _\text{out}$ focuses only on the mapped labels from the source domain, as only these labels relate to the performance of VR.
>
> We hope that our responses can satisfactorily address your concerns. Thank you again for your time to provide us with valuable feedback.

---

> > ### Comment · Reviewer_1Jug · 2024-11-24
> > **Reply to Authors**
> >
> > Thanks for your detailed responses. I still have two questions.
> >
> > **Q1.** Is there any unique or new challenge in the adversarial robustness of visual reprogramming compared to image classification? This can help highlight the significance of your work.
> >
> > **Q2.** What is the role of $f_\text{out}\circ f_\mathcal{S}$ on the domain $\mathcal{S}$ ? To me, this term does not have a exact meaning in the context of visual reprogramming.

---

> > > ### Author Response · Authors · 2024-11-29
> > > **Looking forward to your response**
> > >
> > > We deeply appreciate your constructive suggestions and feedback, and sincerely apologize for any inconvenience caused.
> > >
> > > We have identified and outlined several unique and new challenges in the adversarial robustness of visual reprogramming, which significantly highlight the importance of our work. Moreover, we have carefully revised Definition 1, providing detailed explanations for each component as well as the overall meaning. We have also ensured that these revisions do not affect our conclusions.
> > >
> > > We hope our response will address your concerns and better emphasize our contributions. We would be truly grateful If you could consider raising your rating in recognition of these contributions. Regardless of the final rating, we sincerely respect and appreciate your review process and insightful comments, which have been invaluable in improving our work. Due to the limited time for discussion, we look forward to your response.

---

> > > > ### Comment · Reviewer_1Jug · 2024-11-30
> > > > **Reply to Authors**
> > > >
> > > > Thanks for your timely responses. After the discussion, most of my concerns have been properly addressed. The authors inroduced the issue of adversarial robustness to visual reprogramming and provided a theoretical framework to support their practices. I will raise my rating to 6.
> > > >
> > > > I highly suggest the authors include the discussions above into their revision, especially the comparison with [1] and the potential applications for MLLMs.

---

> > > > > ### Author Response · Authors · 2024-11-30
> > > > > **Thanks for your feedback!**
> > > > >
> > > > > Thank you for recognizing our work. We greatly appreciate your suggestions and feedback, which have significantly improved the quality of our paper. We are currently unable to submit a revised manuscript, but we will definitely incorporate the aforementioned discussion into the final version. Thank you again for your support of our work and your valuable comments!

---

> ### Author Response · Authors · 2024-11-24
> **We appreciate your response**
>
> Thank you for your response. Our responses to your new questions (Q) are provided as follows.
>
> **Q1: Is there any unique or new challenge in the adversarial robustness of visual reprogramming compared to image classification? This can help highlight the significance of your work.**
>
> Compared to image classification, there are unique or new challenges in the adversarial robustness of visual reprogramming as follows:
>
> - **Compatibility Between Adversarial Training Methods and VR Frameworks**: In VR, the trainable components are limited to input transformation $f_\text{in}$ and output label mapping $f_\text{out}$. This restriction presents a unique challenge in designing adversarial training algorithms. For example, when using adversarially pre-trained models, it is a new challenge to simultaneously account for such constraints and better leverage the robustness of pre-trained models to develop a more compatible and efficient adversarial training method.
>
> - **Design of $f_\text{in}$ and $f_\text{out}$ Methods**: Our experiments demonstrate that the choice and design of $f_\text{in}$ and $f_\text{out}$ methods significantly impact the adversarial robustness of the model. Designing $f_\text{in}$ and $f_\text{out}$ methods with inherent consideration for adversarial robustness is an unique challenge in the adversarial robustness of VR.
>
> In our work, we theoretically and experimentally demonstrate that the robustness of VR is influenced by multiple factors: the robustness of the pretrained model, the use of adversarial training, and different input transformations and output label mappings. This lays the foundation for future research to address the challenges mentioned above.
>
> **Q2: What is the role of $f_\text{out}\circ f_\mathcal{S}$ on the domain $\mathcal{S}$ ? To me, this term does not have a exact meaning in the context of visual reprogramming.**
>
> Thanks so much for pointing out the issue of $f_\text{out}\circ f_\mathcal{S}$, which helps us identify an oversight in our theorem. We have carefully reviewed Definition 1 and Theorem 1 and found that $f_\text{out}$ was mistakenly added to the last term. The revised Definition 1 is given as follows:
>
> $$
> d_{f_ \mathrm{out} \circ f_ \mathcal{S} \circ f_ \mathrm{in},\mathcal{F}_ \mathcal{S}}^{\mathrm{adv}, (\rho)}(\mathcal{S},\mathcal{T}) = \sup _{f^{\prime} _\mathcal{S} \in \mathcal{F} _{\mathcal{S}}} \left \vert \mathrm{disp} _{\mathcal{T}}^{\mathrm{adv},(\rho)}(\right.f _\mathrm{out} \circ f _\mathcal{S}^\prime \circ f _\mathrm{in},  f _\mathrm{out} \circ f _\mathcal{S} \circ f _\mathrm{in})  \left. -\mathrm{disp} _{\mathcal{S}}^{\mathrm{adv},(\rho)}(f^{\prime} _\mathcal{S}, f _\mathcal{S}) \right \vert.
> $$
>
> Specifically, the last term is the adversarial margin disparity (AMD) on the source domain, which quantifies the maximum margin disparity between models $f^{\prime} _\mathcal{S}$ and $f _\mathcal{S}$ within the $\epsilon$-perturbation range around the instance $\boldsymbol{x}$, meaning the expectation of inconsistent classification between $f^{\prime} _\mathcal{S}$ and $f _\mathcal{S}$ under $\epsilon$-perturbation in the domain $\mathcal{S}$. The first term is similarly defined on the target domain. The difference between these two terms in Adversarial Margin Disparity Discrepancy of Visual Reprogramming with Adversarial Robustness (AMDD-VRA) is used to measure the adversarial distribution distances between the source domain $\mathcal{S}$ and the target domain $\mathcal{T}$. It can be observed that $f _\text{in}$ and $f _\text{out}$ influence the value of AMDD-VRA. For instance, when the source domain and target domain are completely identical, if both $f _\text{in}$ and $f _\text{out}$ are identity mappings, the AMDD-VRA value is 0. However, in the context of VR settings, where the source and target domains are not identical, $f _\text{in}$ and $f _\text{out}$ can play a role in either narrowing or widening the adversarial discrepancy between the two different domains. Furthermore, based on the definition of AMD, it can be inferred that whether adversarial training is employed for $f _\text{in}$ and $f _\text{out}$ may impact the first term of AMDD-VRA, thereby affecting the overall value of AMDD-VRA.
>
>
> Fortunately, we have thoroughly examined the paper and confirmed that this oversight does not affect our conclusions. We have revised Definition 1 and the proof of Theorem 1 in Appendix A, highlighting the changes in blue, and have uploaded the revised version. Furthermore, the issue ($f_\mathrm{out} \circ f_\mathcal{S}$) addressed in Q2 will no longer be present in the revised manuscript.
>
> Thanks again for the reviewer's carefulness!

---

### Meta-Review · Area_Chair_H6pF · 2024-12-22

**Metareview:**

This paper studies visual reprogramming (VR), focusing on its adversarial robustness aspect. Its key technique contribution lies in demonstrating that 1) leveraging models that have been adversarially trained and 2) applying adversarial training techniques during the reprogramming process can both improve adversarial robustness. A theoretical framework is also provided to establish an adversarial robustness risk upper bound for VR.

Overall, the reviewers find this paper well written and easy to follow, and acknowledge the strong robustness shown in the empirical experiments as well as appreciate the established theoretical upper bound of adversarial risks. But meanwhile, several concerns are shared among reviewers: 1) the novelty of the approach is somewhat limited, as the strategy primarily applies existing techniques from adversarial training and VR; 2) the theoretical contribution seems somewhat disconnected from the practical methods proposed; 3) more datasets and architectures should be included in the ablation studies; and 4) a more comprehensive discussion of recent related works and the limitations of the method should be included.

The authors were actively engaged in the rebuttal period, well addressing the concerns (2)-(4). This led three reviewers to increase their ratings to 6 (marginally above the acceptance threshold). The reviewer xfee is still slightly negative about this paper (rating it as 5), concerning the novelty and the availailibity of adversarially trained models.

Regarding novelty, the AC believes that, while the developed strategy is a straightforward combination of existing techniques, the empirical benefits achieved are significant, and the theoretical bound is a valuable contribution. As for the availability of adversarially trained models, the AC considers this a minor concern, especially given the rapid growth of the open-source community and the increasing interest in trustworthy models.

Based on these considerations, the AC supports the acceptance of this paper.

**Additional Comments On Reviewer Discussion:**

The major concerns are listed in my meta-review. Points (2)–(4) were effectively addressed in the rebuttal, successfully persuading three reviewers to support accepting the submission. The AC believes the unaddressed Point (1) (i.e., the novelty concern) is outweighed by the overall strengths of the paper, such as its strong empirical results and theoretical support.

Therefore, the AC recommends accepting the paper.

---

### Decision · Program_Chairs · 2025-01-22

Accept (Poster)